

# Ocean Acidification trends and Carbonate System dynamics in the North Atlantic Subpolar Gyre during 2009-2019

David Curbelo-Hernández[1], Fiz F. Pérez[2], Melchor González-Dávila[1,*], Sergey V. Gladyshev[3], Aridane G. González[1], David González-Santana[1], Antón Velo[2], Alexey Sokov[3], and J. Magdalena Santana-Casiano[1].

[1] Instituto de Oceanografía y Cambio Global (IOCAG), Universidad de Las Palmas de Gran Canaria (ULPGC). Las Palmas de Gran Canaria, Spain.

[2] Instituto de Investigaciones Marinas (IIM), CSIC, Vigo, Spain.

[3] P. P. Shirshov Institute of Oceanology, Russian Academy of Sciences, Moscow, Russian Federation

*Corresponding Author: Melchor González-Dávila (melchor.gonzalez@ulpgc.es)

**Keypoints:**

During the 2010s, the subpolar North Atlantic experienced a 50-86% increase in anthropogenic $CO_2$, accelerating by <10% the acidification.

Anthropogenic $CO_2$ contributed to acidification by 53-68% in upper layers and >82% in the interior ocean.

The acidification trends (0.0006 and 0.0032 units $yr^{-1}$) declined the $\Omega_{Ca}$ and $\Omega_{Arag}$ by 0.004-0.021 and 0.003-0.0013 units $yr^{-1}$, respectively.





**Abstract**
The $CO_2$-carbonate system dynamics in the North Atlantic Subpolar Gyre (NASPG) were
evaluated between 2009 and 2019. Data was collected aboard eight summer cruises through
the CLIVAR 59.5ºN section. The Ocean Acidification (OA) patterns and the reduction in the
saturation state of calcite ($\Omega_{Ca}$) and aragonite ($\Omega_{Arag}$) in response to the increasing
anthropogenic $CO_2$ ($C_{ant}$) were assessed within the Irminger, Iceland and Rockall basins
during a poorly-assessed decade in which the physical patterns reversed in comparison with
previous well-known periods. The observed cooling, freshening and enhanced ventilation
increased the interannual rate of accumulation of $C_{ant}$ in the interior ocean by 50-86% and
the OA rates by close to 10%. The OA trends were 0.0013-0.0032 units yr$^{-1}$ in the Irminger
and Iceland basin and 0.0006-0.0024 units yr$^{-1}$ in the Rockall Trough, causing a decline in
$\Omega_{Ca}$ and $\Omega_{Arag}$ of 0.004-0.021 and 0.003-0.0013 units yr$^{-1}$, respectively. The $C_{ant}$-driven rise
in total inorganic carbon ($C_T$) was the main driver of the OA (contributed by 53-68% in upper
layers and >82% toward the interior ocean) and the reduction in $\Omega_{Ca}$ and $\Omega_{Arag}$ (>64%). The
transient decrease in temperature, salinity and $A_T$ collectively counteracts the $C_T$-driven
acidification by 45-85% in the upper layers and in the shallow Rockall Trough and by <10%
in the interior ocean. The present investigation reports the acceleration of the OA within the
NASPG and expands knowledge about the future state of the ocean.
**Keywords:** Ocean Acidification, Anthropogenic Carbon, North Atlantic Subpolar Gyre.

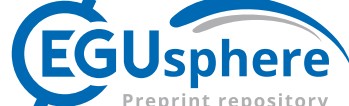

## 1. Introduction

The ocean uptake of approximately one-third of the $CO_2$ released into the atmosphere (Friedlingstein et al., 2023; Gruber et al., 2019a) has an important role in the climate regulation causing changes in the marine carbonate chemistry. The exponential increase in the global ocean $CO_2$ sink in phase with those of anthropogenic emissions (Friedlingstein et al., 2023) has resulted in a long-term decrease in the concentration of carbonate ions ($[CO_3^{2-}]$) and pH. This process has been collectively referred to as Ocean Acidification (OA; Caldeira and Wickett, 2005, 2003; Doney et al., 2009; Orr et al., 2005; Raven et al., 2005; Feely et al., 2009) and favour the dissolution of calcium carbonate ($CaCO_3$). It affects not only calcifying marine organisms and ecosystems which use the biogenic $CaCO_3$ forms of calcite and aragonite (e. g. Gattuso et al., 2015; Langdon et al., 2000; Pörtner et al., 2004, 2019; Riebesell et al., 2000) but also the global biogeochemical cycles (Gehlen et al., 2011; Matear and Lenton, 2014).

The absorption of anthropogenic $CO_2$ has reduced the pH of the global surface ocean by 0.1 units since preindustrial times, representing approximately a 30% increase in acidity (Caldeira and Wickett, 2003). The model projections estimate that the pH could fall by 0.5 units by the end of the century if global $CO_2$ emissions continue to rise, while a drop of 0.2 units is expected for the most conservative scenario (Caldeira and Wickett, 2005; Orr et al., 2005, 2011; Raven et al., 2005). However, as the absorption and storing of anthropogenic carbon ($C_{ant}$) within the ocean is not uniform (Sabine et al., 2004a), OA rates may show a significant spatial variability and should be regionally studied. The temporal evolution of the carbonate system variables in surface waters are monitored and assessed in several time-series stations located across different ocean regions (Bates et al., 2014). The largest OA rates are expected to occur across high northern and southern latitudes (Bellerby et al., 2005; Orr et al., 2005), where deep convective overturning and subduction occur favouring the entrance of $C_{ant}$ in the interior ocean (Maier-Reimer and Hasselmann, 1987; Lazier et al., 2002; Sarmiento et al., 1992).

The North Atlantic is one of the strongest $CO_2$ sinks and stores over 25% of the $C_{ant}$ accumulated in the global ocean (e. g. Gruber et al., 2019; Khatiwala et al., 2013; Pérez et al., 2024, 2010, 2008, 2024; Sabine et al., 2004; Takahashi et al., 2009). The Atlantic Meridional Overturning Circulation (AMOC) plays a significant role by conveying acidified $C_{ant}$-loaded waters polewards and exporting them to the ocean interior across deep-water formation areas

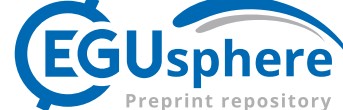

(Lazier et al., 2002; Pérez et al., 2013, 2008; Steinfeldt et al., 2009). It contributes to
homogenize the $C_{ant}$ and pH in the whole water column in such regions and exported these
properties southwards to the global deep ocean (Perez et al., 2018). Thus, the North Atlantic
behaves as a crucial region for understanding the impacts of anthropogenic forcing on the
global ocean.
OA has been widely studied in the North Atlantic through the monitoring of the ocean
physicochemical properties at time-series stations (summarized by Bates et al., 2014) placed
in subtropical and subpolar latitudes: the European Station for Time series in the Ocean at the
Canary Islands (ESTOC; 29.04ºN, 15.50ºW; González-Dávila et al., 2010; González-Dávila
and Santana-Casiano, 2023; Santana-Casiano et al., 2007), the Bermuda Atlantic Time-series
Study (BATS; 32.0ºN, 64.0ºW; Bates et al., 2012), the Irminger Sea Time Series (IRM-TS;
64.3ºN, 28.0ºW; Olafsson et al., 2010) and the Iceland Sea Time Series (IS-TS; 68.0ºN,
12.66ºW; Olafsson et al., 2009, 2010). OA rates has also been evaluated along transects
through repeated hydrographic cruises (i.e. Guallart et al., 2015; García-Ibáñez et al., 2016;
Vázquez-Rodríguez et al., 2012b) or even covered by volunteer observing ships (Fröb et al.,
2019). These investigations have revealed a rate of decrease in pH of ~0.001-0.002 units $yr^{-1}$.
Moreover, González-Dávila and Santana-Casiano, (2023) has recently indicated that these
rates are increasing since 1995.
The assessment of OA is of especial interest across the North Atlantic Subpolar Gyre (NASPG;
50-60ºN), where the atmospheric $CO_2$ sink is particularly strong and the deep-water formation
processes favour the storage of $C_{ant}$ through the whole water column (Gruber et al., 2019b;
Sabine et al., 2004b; Watson et al., 2009, Pérez et al. 2008). Likewise, the deep-water
formation processes create the largest and deepest ocean environments supersaturated for
aragonite (at more than 2000 m depth; Feely et al., 2004; Jiang et al., 2015), which is the main
$CaCO_3$ mineral for Cold-water corals (CWC; Roberts et al., 2009) and some pteropods
(Bathmann et al., 1991; Urban-Rich et al., 2001). These deep biomes are predicted to be one
of the first in the global ocean affected by OA, mainly due to the shoaling of the Aragonite
Saturation Horizon and its progressive exposition to undersaturated conditions for aragonite at
intermediate and deep waters (Raven et al., 2005; Guinotte et al., 2006; Turley et al., 2007;
Roberts et al., 2009).





The physical processes along the NASPG, which are subject to significant spatiotemporal
variability introduced by the atmospheric forcing and climatology on an interannual scale,
directly influenced the biogeochemistry (Corbière et al., 2007; Fröb et al., 2019). The changes
in North Atlantic Current (NAC) modifies the poleward heat transport from subtropical
latitudes and the air-sea interactions, influencing temperature patterns (Josey et al., 2018;
Mercier et al., 2015). Recent studies noticed the surface cooling and freshening of the NASPG
in the 2010s (Holliday et al., 2020; Josey et al., 2018; Robson et al., 2016; Tesdal et al., 2018)
contrasting with the period of warming and salinification in the 1990s extended until 2005
(Häkkinen and Rhines, 2004; Hátún et al., 2005; Robson et al., 2014). Anomalously heat loss
and winter deep convection were found to be of high intense since 2008 contributing to the
extreme cold anomaly along the NASPG (e. g. De Jong et al., 2012; de Jong and de Steur,
2016; Fröb et al., 2019, 2016; Gladyshev et al., 2016b, 2016a; Piron et al., 2017; Våge et al.,
2009). These fluctuations in the vertical mixing and ocean circulation patterns introduces
changes in the distribution of the carbonate system variables.
Several investigations have evaluated the drivers, trends and impacts of OA in the western
NASPG at the Irminger and Iceland basins (e. g. Fontela et al., 2020; García-Ibáñez et al.,
2021, 2016; Perez et al., 2018; Pérez et al., 2021; Ríos et al., 2015), while few studies have
addressed it along the Rockall Trough (e. g. McGrath et al., 2013, 2012a, 2012b, Humphreys
et al., 2016) due to lack of repeated hydrographic sections or time-series stations and
subsequent limitation of continuous surface-to-bottom data. The high longitudinal variability
in the NASPG caused by the influence of different circulation patterns and water masses
(García-Ibáñez et al., 2018, 2015) introduced several physicochemical heterogeneities
between the Irminger and Iceland with the Rockall basin (Ellett et al., 1986; McGrath et al.,
2013, 2012b; Holliday et al., 2000). These differences in the distributions of Marine Carbonate
System (MCS) variables should be considered to improve our understanding of OA in the
entire North Atlantic.
This study evaluated the OA in the NASPG across the Irminger, Iceland and Rockall basins
during the 2010s. High-quality direct measurements of $CO_2$ system variables from eight
hydrographic cruises occupying 59.5ºN between 2009 and 2019 were used to evaluate the
drivers and trends of pH, and the potential effects of OA on calcifying organisms of changes



in calcite (ΩCa) and aragonite (ΩAr) saturation states. This study advances our understanding
of the complexities associated with OA in the NASPG and supports ongoing efforts to model
and predict future acidification scenarios in the North Atlantic and global ocean.
**2.   Methodology**
**2.1.    Data collection**
Data was collected along the hydrographic CLIVAR 59.5ºN section (Daniault et al., 2016;
Gladyshev et al., 2016c, 2018, 2017; Sarafanov et al., 2018) from 8 summer cruises with dates
spanning 11 years (2009-2019). The section covers the length of the North Atlantic at 59.5ºN
between Scotland and Greenland (4.5-43.0ºW), crossing the Irminger and Iceland basins and
the Rockall Trough (Figure 1). Generally, the sampling stations were equidistantly spaced
every 20 nmi apart (~1/3º longitude) and repeated in all the cruises except for the cruise of
2016, when the station spacing was decreased to 10 nmi over Reykjanes Ridge western and
eastern slopes. The distance between stations over the east Greenland slope and shelf always
decreased from 10 nmi to about 2 nmi. The surface-to-bottom sampling and in situ
measurements were performed by using a SBE 911plus CTD with SBE32 Carousel containing
24 Niskin bottles (10 L) with additional sensors for pressure (P), dual temperature (T) and
salinity (S), and dissolved oxygen (DO). The eight cruises included in the new dataset are the
result of an international collaboration between researchers from the P. P. Shirshov Institute of
Oceanology at the Russian Academy of Science and the Marine Chemistry research group
from the Oceanography and Global Change Institute (QUIMA-IOCAG) at the University of
Las Palmas de Gran Canaria (ULPGC). A detailed overview and metadata of the cruises is
given in Table 1.
**2.1.1.  CO₂ system variables measurements**
The analysis of the MCS variables followed the same analytical methodology and provided
high-quality $CO_2$ measurements in all the hydrographic cruises. It includes the sampling and
data collection techniques, quality control and calculation procedures published in the updated
version of the DOE method manual for $CO_2$ analysis in seawater given by Dickson et al., 2007.
The seawater samples were onboard analysed for total alkalinity ($A_T$) and total inorganic
carbon ($C_T$) determination by using a VINDTA 3C and following Mintrop et al., (2000). The
$A_T$ was analysed by potentiometric titration with HCl to the carbonic acid endpoint and



determined through the developing of the full titration curve (Millero et al., 1993; Dickson and
Goyet, 1994). The $C_T$ was determined through coulometric titration (Johnson et al., 1993). The
VINDTA 3C was *in situ* calibrated through the titration of Certified Reference Material
(CRMs; provided by A. Dickson at Scripps Institution of Oceanography), giving values with
an accuracy of ±1.5 μmol kg$^{-1}$ for $A_T$ and ±1.0 μmol kg$^{-1}$ for $C_T$.
Spectrophotometric pH measurements (Clayton and Byrne, 1993) in total scale at constant
temperature of 15ºC (pH$_{T,15}$) were performed for the cruises between 2009 and 2016. A
spectrophotometric pH sensor (SP101-SM) developed by the QUIMA-IOCAG group at the
ULPGC in collaboration with SensorLab (González-Dávila, 2014; González-Dávila et al.,
2016) was used. The method uses 4 wavelengths analysis for the m-cresol purple, includes
auto-cleaning steps and performs a blank for pH calculation immediately after the dye
injection. The spectrophotometric sensor was *in situ* tested by using a TRIS seawater buffer
and provided pH$_{T15}$ values with an accuracy of ±0.002 units. However, DelValls and Dickson,
(1998) reported an uncertainty of the spectrophotometric pH determination associated to the
TRIS used for calibration of -0.0047 units. Hence, the experimental pH values were corrected
by adding 0.0047 units.
### 2.1.2. Dissolved oxygen (DO) measurements
The WINKLER method introduced by Winkler (1888) and optimized by Carpenter (1965) and
Carrit and Carpenter (1966) was used to analytically determine the dissolved oxygen (DO) of
the seawater samples in all the cruises from 2009 to 2016. The seawater samples for DO
determination were collected from the bottle samples in pre-calibrated glass wide-neck bottles
avoiding bubble formation. The temperature of the water was recorded during the sampling.
All the reagents and solutions used for dissolved oxygen determination were prepared
following the procedures described by Dickson and Goyet (1994) and their possible impurities
were controlled by determining a blank every 2 days.
As DO could not be analytically measured during the cruise of 2019 (due to limitations
related with the oceanographic cruise plan), it was computed for this year by comparing the
performance of the DO sensor during the cruise of 2019 versus (1) DO data estimated by a
neural network for the cruises of 2016 and 2019 and (2) WINKLER-measured DO data in
the cruise of 2016. The neural network ESPER_NN (Empirical Seawater Property Estimation



Routine) introduced by Carter et al., (2021) was used for DO estimations. The computational
procedure is detailed in Appendix A.

### 2.2.   Data processing

#### 2.2.1.   Evaluation of the internal consistency of the data using CANYON-B

The measured and determined data were compared with estimations given by the Bayesian
neural network "CANYON-B" (Bittig et al., 2018), a re-developed and more robust neural
network based on CANYON (CArbonate system and Nutrients concentration from
hYdrological properties and Oxygen using a Neural-network; Sauzède et al., 2017).
CANYON-B estimates the four MCS variables ($A_T$, $C_T$, $pH_T$ and $pCO_2$) and macronutrients
concentrations ($PO_4^{3-}$, $NO_3^-$ and $Si(OH)_4$, hereinafter $PO_4$, $NO_3$ and $Si(OH)_4$) as a function of
a simple set of input variables which include P, T, S, DO, latitude, longitude and date. This
neural network is trained on and validated against bottle and sensor data from GLODAPv2,
GO-SHIP and Argo profiles, and provides a local uncertainty for each variable. The standard
errors of estimate reported for CANYON-B by Bittig et al., (2018) are 6.3 µmol kg$^{-1}$ for $A_T$,
7.1 µmol kg$^{-1}$ for $C_T$, 0.013 units for pH, 20 µatm for $pCO_2$, 0.051 µmol kg$^{-1}$ for $PO_4$, 0.68
µmol kg$^{-1}$ for $NO_3$ and 2.3 µmol kg$^{-1}$ for $Si(OH)_4$. The crossover analysis between measured
and estimated data did not show systematic differences but individual outliers. The measured
data that were higher/lower than the CANYON-B estimate by plus/minus twice the predicted
variable uncertainty of the neural network were considered as outliers and removed from the
dataset.
The total amount of measured data was 8974 for $A_T$, 7495 for $C_T$, 8706 for $pH_T$, 9656 for DO,
9114 for $PO_4$ and 9192 for $Si(OH)_4$. The difference between the measured and CANYON-B-
estimated variables (referred hereinafter as canyon-estimated variables) were performed for
each sample in which CANYON-B could be applied (samples with availability of T, S and DO
measurements). The number of data, mean values and standard deviation of the measured
variables for each cruise were summarized in Table S1. The average differences with the 95%
confidence interval for each cruise are shown in Table S2. The average differences for the
entire period (2009-2019) were lower than 2.1 µmol kg$^{-1}$ for $A_T$, 2 µmol kg$^{-1}$ for $C_T$, 0.0002 for
$pH_T$, 0.02 µmol kg$^{-1}$ for PO4 and 0.25 µmol kg$^{-1}$ for $Si(OH)4$.

#### 2.2.2.   Computational methods



The computational procedures to calculate MCS system variables applied in this investigation
used the $CO_2SYS$ programme developed by Lewis and Wallace, (1998) and run with the
MATLAB software (van Heuven et al., 2011; Orr et al., 2018; Sharp et al., 2023). The set of
constants used for computations includes the carbonic acid dissociation constants of Lueker et
al., (2000), the $HSO_4^-$ dissociation constant of Dickson, (1990), the HF dissociation constant
of Perez and Fraga, (1987) and the value of $[B]_T$ determined by Lee et al., (2010). The pH in
total scale at *in situ* temperature ($pH_T$) was computed from the measured $A_T$ and $pH_{T,15}$ (the
computed $C_T$ was given as an output). The $pH_T$ for the cruise of 2019, in which direct pH
measurements were not performed, was computed from the measured $A_T$ and $C_T$.
In addition, as three of the four MCS variables were measured in the rest of the cruises and
due to gaps in data, an intercomparison between measured and computed $C_T$ and $pH_{T15}$ was
performed. It considers the availability of measurements for each latitude, longitude and time
and the differences between the measured and computed pH with the canyon-estimated $pH_T$.
The use of measured or computed $C_T$ followed these conditions: (1) If there is measured $C_T$
but not measured pH, measured $C_T$ was used, (2) if there is measured pH but not measured $C_T$,
computed $C_T$ was used, (3) and if there is measured $C_T$ and pH, measured $C_T$ was used when
the differences between measured and canyon $pH_T$ is lower than the differences between
computed and canyon-estimated $pH_T$, while computed $C_T$ was used when the opposite
happens. In total, 6375 measured and 2872 computed $C_T$ data were used in this study (69%
and 31%, respectively). The average differences in each cruise between the combined
(measured and computed, also referred as "$C_{T (new)}$") and canyon-estimated $C_T$ variable is
provided in Table S2. The amount and percentage of measured and computed $C_T$ data per
cruise is given in Table S3. As the measured $C_T$ was in average 1.9 µmol kg$^{-1}$ higher than the
canyon-estimated and the computed $C_T$ was in average 1.7 µmol kg$^{-1}$ lower, the new
compilation based on these previous conditions allowed to reduce the difference to 1.5 µmol
kg$^{-1}$.

### 2.2.3. Anthropogenic $CO_2$ ($C_{ant}$) calculation

The anthropogenic $CO_2$ ($C_{ant}$) was estimated by using the biogeochemical back-calculation
$\phi C_T^o$ method, which has an overall estimated uncertainty of ±5.2 µmol kg$^{-1}$ (Pérez et al., 2008;
Vázquez-Rodríguez et al., 2009). The method considers the change of $C_T$ between the



preindustrial era (1750) and the time of the observations, as well as the processes involved in
the uptake and distribution of $C_{ant}$ (biogeochemistry, mixing processes and air-sea fluxes). The
$C_{ant}$ was calculated (Eq. 1) as the difference between the $C_T$ at the time of observation, the $C_T$
that the seawater would have in equilibrium with a preindustrial atmosphere (preformed $C_T$;
$C_T^{pre}$), the offsets of such equilibrium values (air-sea $CO_2$ disequilibrium; $\Delta C_T^{dis}$) and the
changes in $C_T$ due to the organic and carbonate pumps ($\Delta C_T^{bio}$). The $C_T$ and $A_T$ at the time of
observations and the preformed $A_T$ ($A_T^0$) are needed as input parameters and the computational
procedure was described by Vázquez-Rodríguez et al., (2012).

$$C_{ant} = C_T - C_T^{pre} - \Delta C_T^{dis} - \Delta C_T^{bio} \tag{1}$$


The $\phi C_T^o$ method is an improved process-based $C_{ant}$ estimation method tested and widely
applied in the Atlantic Ocean (Vázquez-Rodríguez et al., 2009) which present distinctive
characteristics relative to existing $C_{ant}$ approaches, such as the classical $\Delta C^*$ (GSS' 96; Gruber
et al., 1996) and the TrOCA (Touratier et al., 2007). The main advantages of the $\phi C_T^o$ method
has been described by Pérez et al., (2008).
**2.2.4. Water mass characterization**
The characterization of the basins and water masses was done by considering the 2006-2021
mean combined CLIVAR 59.5ºN section constructed with potential vorticity, dissolved oxygen
and salinity together with the large-scale circulation in the North Atlantic (e. g. Lherminier et
al., 2010; Pérez et al., 2021; Sarafanov et al., 2012; Schmitz and McCartney, 1993; Schott and
Brandt, 2007; Sutherland and Pickart, 2008). A schematic diagram with the main surface and
deep currents in the NASPG is depicted in Figure 1a. The basin division considered the NAC
pathways and revealed a west-to-east distribution comprising the Irminger and Iceland basins
and the Rockall Trough. The Iceland basin was delimited along its eastern boundary by the
central NAC branches around the northern part of the Haton Bank and George Bligh Bank,
and along its western boundary by the Return Current over the eastern flank of the Reykjanes
Ridge slope. This suggest that the Iceland basin could be longitudinally separated in two
subregions: the western Iceland basin (24.0-29.5ºW) and the eastern Iceland basin (14.0-
24.0ºW).



The upper layers were mainly occupied by Subpolar Mode Waters (SPMW) and North Atlantic
Central Waters (NACW). SPMW is formed in the Iceland basin (McCartney and Talley, 1982;
Brambilla and Talley, 2008; Tsuchiya et al., 1992; Van Aken and Becker, 1996), flow eastward
to the Rockall Trough and recirculate across the Reykjanes Ridge (Brambilla and Talley, 2008).
In the Irminger basin, SPMW flow with the Irminger Current to the north over the western
Reykjanes Ridge flank and to the south over the eastern Greenland slope (Figure 1a). Thus,
SPMW signal was detected in the western and eastern Irminger basin up to 400-700 m depth
and limited to subsurface depths in the central part of the basin. NACW were placed above
SPMW east of the Irminger basin and separated in two branches: Eastern North Atlantic
Central Water (ENACW), formed by winter convection in the intergyre region and moved
poleward from the Bay of Biscay through the Rockall Trough (Harvey, 1982; Pollard et al.,
1996), and Western North Atlantic Central Water (WNACW), flowing northward with the
NAC along the western Iceland basin. The intermediate layers were mainly occupied by
Labrador Sea Water (LSW), formed in the Labrador Sea and transported eastward (e. g. Pickart
et al., 2003; Fröb et al., 2016). LSW path diverges into two cores when it reaches the Reykjanes
Ridge (Álvarez et al., 2004; Pickart et al., 2003): a fraction of LSW rapidly moved to the
Irminger basin and incorporated into the Deep Western Boundary Current (DWBC) (Bersch et
al., 2007) and a second LSW core was transported eastward into the Iceland and Rockall
basins. In the Irminger and western Iceland basin, LSW placed above Iceland-Scotland
Overflow Water (ISOW), which originated from the overflow of Norwegian Sea waters over
the Iceland–Scotland ridges and flowed southward and below 1500 m depth through the
western NASPG (van Aken and de Boer, 1995; Dickson et al., 2002; Fogelqvist et al., 2003).
The bottom of the western Irminger basin was occupied by Denmark Strait Overflow Water
(DSOW), recently formed from deep waters from the Nordic seas flowing southward over the
Greenland-Iceland ridge and sinking through the eastern Greenland slope (Read, 2000;
Stramma et al., 2004; Yashayaev and Dickson, 2008). LSW core transported eastwards rises
in depth through the western Haton Bank flank and occupy the bottom depths in the eastern
Iceland basin and in the Rockall Trough. A low-ventilated thermocline layer is placed between
SPMW and LSW in the eastern NASPG (García-Ibáñez et al., 2016), which represent the
product of mixing with waters coming from the south (i. e. Mediterranean Waters; MW).



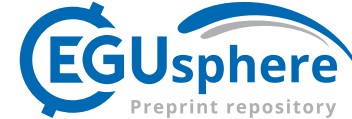

The physical and biogeochemical interannual changes were analysed in the main basins and water masses. In order to enhance the comprehension of the spatial distribution and trends of the biogeochemical variables and to facilitate comparisons with previous studies along the NASPG, the hydrographic characterization was simplified based on the following principles: (1) the Iceland basin was not divided into its western and eastern parts and its longitudinal span was delimited by the Reykjanes Ridge (29.5ºW) and the Haton Bank (17ºW), (2) upper Labrador Sea Water (uLSW) was separated from deeper LSW (e. g. Stramma et al., 2004), (3) the weak and spatially-limited influence of the return current and WNACW was removed by considering the upper and intermediate layers of both the Irminger and Iceland basin fully occupied by SPMW above uLSW, and (4) only the east branch of NACW (ENACW), placed above SPMW, was contemplated for the upper Rockall Trough.

The whole water column was separated in layers delimited by potential density isopycnals at a reference pressure of 0 dbar following Azetsu-Scott et al. (2003), Kieke et al. (2007), Pérez et al. (2008) and Yashayaev et al. (2008). The vertically distributed water masses separated in density layers is represented for the entire section in Figure 1b. The vertical characterization in density layers allows to consistently compare the low-variable physical and chemical properties within each water mass, enabling to assume linearity in the ocean $CO_2$ system. The determination of the isopycnal limits between layers in the Irminger and Iceland basins followed previous biogeochemical studies in the western boundary of the North Atlantic (Fontela et al., 2020; García-Ibáñez et al., 2016; Pérez et al., 2010, 2008; Vázquez-Rodríguez et al., 2012a). The surface-to-bottom distribution of the main water masses in these basins (with their respective $\sigma_0$ lower limits shown in brackets) was SPMW (27.68 kg m$^{-3}$), uLSW (27.76 kg m$^{-3}$), LSW (27.81 kg m$^{-3}$) and ISOW (27.88 kg m$^{-3}$). The low temperature and salinity DSOW were considered at the bottom of the westernmost part of the Irminger basin. The hydrography of the Rockall Trough has been characterized in previous studies in the Northeast Atlantic (e. g. Ellett et al., 1986; Harvey, 1982; McGrath et al., 2012a, 2012b; Holliday et al., 2000). The considered surface-to-bottom distribution of the main water masses was ENACW (27.35 kg m$^{-3}$), SPMW (27.68 kg m$^{-3}$) and LSW (bottom).

### 2.2.5.  pH$_T$ trends deconvolution



OA trends arise due to the combined variations in T, S, $C_T$ and $A_T$. The influence of each driver
on OA was analysed by assuming linearity and employing a first-order Taylor-series
deconvolution to evaluate the $pH_T$ trends (Fröb et al., 2019; García-Ibáñez et al., 2016; Pérez
et al., 2021; Takahashi et al., 1993; Tjiputra et al., 2014). Partial derivatives of $pH_T$ versus T,
S, $C_T$ and $A_T$ were calculated based on mean properties of each layer by using the most recent
equation (Eq. 2) given by Pérez et al., (2021). This equation introduced salinity-normalized $C_T$
and $A_T$ ($NX_T = X_T/S*35$) to remove the effect of the freshwater fluxes in the variation of $A_T$
and $C_T$.
$$\frac{dpH_T}{dt} = \frac{\partial pH_T}{\partial T}\frac{dT}{dt} + \left(\frac{\partial pH_T}{\partial S} + \frac{NC_T}{S_0}\frac{\partial pH_T}{\partial C_T} + \frac{NA_T}{S_0}\frac{\partial pH_T}{\partial A_T}\right)\frac{dS}{dt} + \frac{S}{S_0}\frac{\partial pH_T}{\partial C_T}\frac{dNC_T}{dt} + \frac{S}{S_0}\frac{\partial pH_T}{\partial A_T}\frac{dNA_T}{dt}$$ (2)
It is important to remark that the changes in $NA_T$ and $NC_T$ are linked with biogeochemical
processes which have different influences: the processes involved in the organic carbon pump
contribute to strongly change the $NC_T$ weakly affecting the $NA_T$, while those involved in the
carbonate pump affect the $NA_T$ twice as much as $NC_T$. The complexity and heterogeneity of
the processes that govern the $pH_T$ change were considered by this equation.
**2.2.6. Calculation of the state of saturation of Calcite ($\Omega_{Ca}$) and Aragonite ($\Omega_{Arag}$):**
**trends and drivers**
The adverse impacts of OA on marine calcification processes and its correlation with the
saturation states of Calcite ($\Omega_{Ca}$) and Aragonite ($\Omega_{Arag}$) has been commonly demonstrated (e.
g. Gattuso et al., 2015; Langdon et al., 2000; Pörtner et al., 2004, 2019; Riebesell et al., 2000).
The $\Omega_{Ca}$ and $\Omega_{Arag}$ were calculated as the product of the ion concentrations of calcium ($[Ca^{2+}]$)
and carbonate ($[CO_3^{2-}]$) divided by the stoichiometry solubility products ($K_{sp}$) for calcite ($K_{Ca}$)
and aragonite ($K_{Arag}$) given by Mucci (1983) (Eq. 3 and 4). The $\Omega_{Ca}$ and $\Omega_{Arag}$ were calculated
with the CO2SYS programme (Lewis and Wallace, 1998) for MATLAB (van Heuven et al.,
2011; Orr et al., 2018; Sharp et al., 2023), applying the set of constants detailed in section

359 2.2.2.

$$\Omega_{Ca} = \frac{[Ca^{2+}][CO_3^{2-}]}{K_{Ca}}$$ (3)
$$\Omega_{Arag} = \frac{[Ca^{2+}][CO_3^{2-}]}{K_{Arag}}$$ (4)



The collective temporal changes in the physico-chemical properties governing the OA
influenced the $\Omega_{Ca}$ and $\Omega_{Arag}$ variations and were considered in this study. The influence of the
potential drivers was analysed by employing a first-order Taylor-series deconvolution to
evaluate the $\Omega_{Ca}$ and $\Omega_{Arag}$ trends, as done with the $pH_T$ (section 2.2.5). Likewise, the
interannual changes of $\Omega_{Ca}$ and $\Omega_{Arag}$ were assumed linear and given by the sum of the partial
derivates of $\Omega_{Ca}$ and $\Omega_{Arag}$ versus each driver (García-Ibáñez et al., 2021) in Eq. 5.
$$\frac{d\Omega}{dt} = \frac{\partial\Omega}{\partial T}\frac{dT}{dt} + \left(\frac{\partial\Omega}{\partial S} + \frac{NC_T}{S_0}\frac{\partial\Omega}{\partial C_T} + \frac{NA_T}{S_0}\frac{\partial\Omega}{\partial A_T}\right)\frac{dS}{dt} + \frac{S}{S_0}\frac{\partial\Omega}{\partial C_T}\frac{dNC_T}{dt} + \frac{S}{S_0}\frac{\partial\Omega}{\partial A_T}\frac{dNA_T}{dt} \qquad (5)$$

**3. Results**
**3.1.    Physicochemical characterization of the water column**
The vertical distribution of the physical and biogeochemical variables is depicted in Figures 2,
3, S2 and S3. These figures exhibited the changes in the water-column properties throughout
the section between 2009 and 2016. The subsurface layers were characterized by warmer and
saltier waters than intermediate and deep layers among the three basins (Figure 2a and 2b). A
West-to-East increase in temperature and salinity throughout the water column was observed
in all the cruises. The temperature and salinity signals were highest in the Rockall Trough (4.5-
11.0ºC and 35.0-35.4, respectively), followed by the Iceland basin (3.0-7.5ºC and 34.9-35.2,
respectively) and the Irminger basin (1.5-6.5ºC and 34.8-35.1, respectively). The longitudinal
differences in temperature were more remarkable toward the upper layers through the SPMW
and uLSW.
The spatial variability in the physical properties introduced heterogeneities in the distribution
of the $CO_2$ system variables. The $A_T$ show a well-correlated direct relationship with salinity
throughout the section ($r^2$=0.89), with lower and vertically-homogenized average values in the
Irminger basin (2302.8-2307.3 µmol kg$^{-1}$ in subsurface waters and 2298.8-2301.0 µmol kg$^{-1}$
in bottom waters) and Iceland basin (2308.7-2315.0 µmol kg$^{-1}$ in subsurface waters and
2305.2-2308.0 µmol kg$^{-1}$ in bottom waters) compared to the Rockall Trough (2317.9-2329.1
µmol kg$^{-1}$ in subsurface waters and 2308.5-2310.9 µmol kg$^{-1}$ in bottom waters). The upper
layers were characterized by low $C_T$ values (2153.7-2160.8 µmol kg$^{-1}$ at the Irminger basin,
2158.1-2168.4 µmol kg$^{-1}$ at the Iceland basin and 2120.1-2131.0 µmol kg$^{-1}$ at the Rockall
Trough), while a rapidly increment with depth was found below 100-200 m depth (2154.7-





2171.2 µmol kg$^{-1}$ throughout the section). The notable difference in the distribution of $A_T$ and
$C_T$ (Figure 2c and 3a, respectively) compared to those of $NA_T$ and $NC_T$ (Figure S2) elucidated
the remarkable significance of freshwater fluxes on the carbon variables fluctuations during
the period of study. The entrance of $C_{ant}$ through the atmosphere-seawater interface caused
higher $C_{ant}$ values in the upper layers (higher than 50 µmol kg$^{-1}$ in the first 1000 m depth;
Figure 3b). The natural component of the $C_T$ ($C_{nat}=C_T-C_{ant}$; Figure 3c) correlated with $C_T$
($r^2$=0.87), and show a distribution characterized by low surface (<2110 µmol kg$^{-1}$) and high
bottom concentrations (>2130 µmol kg$^{-1}$).
The pH$_T$ (Figure 2d) rapidly decreased with depth showing the effect of biological uptake in
the upper layers and remineralization in deeper areas. The subsurface layer up to 100-200 m
depth exhibited pH$_T$ values higher than 8.025 units, which fell to 7.975 units at the bottom
layers. The pH$_T$ profiles reported an intrusion of remineralized and poorly oxygenated water
between 500 and 1000 m depth with relatively low pH$_T$ (<7.975) compared to adjacent layers
in the Iceland basin and in the western part of the Rockall Trough. This thermocline layer was
previously observed at ~500 m depth by García-Ibáñez et al., (2016) along a more meridional
transect which crossed the Iceland basin northwest-southeast. It introduces differences in the
intermediate water masses between the Iceland and Rockall basins with the Irminger basin.
The spatial and interannual fluctuations in the ventilation rates through changes in the water
mass formation and respiration processes represent a source of variability in the
biogeochemical patterns. The apparent oxygen utilization (AOU), defined as the difference
between saturated oxygen (calculated following Benson and Krause, 1984)  and measured
oxygen, was used to assess the ventilation of the water masses (Figure 2e). The high AOU
values indicate low ventilation, while low AOU values indicate the opposite. The slow renewal
of waters with high AOU favour the accumulation of the product of remineralization (de la
Paz et al. 2017). Thus, the areas with higher AOU (Figure 2e) were found to have high
concentration of $C_T$ and low pH$_T$ (Figures 3a and 2d, respectively). The near surface waters
permanently in contact with the atmosphere exhibited the lowest AOU values (<20 µmol kg$^{-1}$
$^1$). The Irminger Basin presents the most significant water column ventilation among the entire
section, with maximum AOU ranging from 35 to 50 µmol kg$^{-1}$ at the LSW and ISOW and the
remarkable intrusion of oxygenated DSOW (>260 µmol kg$^{-1}$ DO) over the continental slope





with AOU ranging from 30 to 40 µmol kg$^{-1}$. The intermediate and deep layers of the Iceland
and Rockall basins were less ventilated, with AOU values higher than 45-50 µmol kg$^{-1}$. The
thermocline layer placed between 500 and 1000 m depth along these two basins presented the
highest maximum AOU throughout the period (>60 µmol kg$^{-1}$). The stagnation of these waters
corresponds with the high $C_T$ and low pH$_T$ (Figures 3a and 2d, respectively) encountered at
intermediate depths and should be considered in its temporal evolution.

### 3.2. Temporal evolution of the physicochemical properties

The interannual trend in the distribution of the physicochemical properties was analysed
through the whole water column across the Irminger, Iceland and Rockall basins by yearly
averaging the variables for each layer, following previous studies in the NASPG (e.g. Fontela
et al., 2020; García-Ibáñez et al., 2016) and applying linear regressions, where the ratios of
interannual change were given by the values of the slopes. The temporal distribution and trends
of the average physicochemical properties (Figures 4, 5, 6, S4, S5 and S6) revealed remarkable
heterogeneities in their interannual evolution within the period 2009-2019 among the different
basins and water masses. The mean properties were represented with error bars that are two
times the error of the mean ($2\sigma = 2 * (Standard\ Deviation/\sqrt{n})$, where $n$ is the number of
bottle samples in each layer and cruise. The interannual ratios are presented along with their
respective standard error of estimate and correlation factors (r$^2$ and p-value) in Table 3 and S4.
The standard errors of the slopes were calculated by considering the standard error of the
annual mean values. The p-values ≤ 0.01 indicated that the trends were statistically significant
at the 99% confidence level, the p-values ≤ 0.05 indicated that the trends were statistically
significant at the 95% confidence level and the p-values ≤ 0.1 indicated that the trends were
statistically significant at the 90% level. Trends with p-values > 0.1 were considered as not
statistically significant but provided an estimation of the temporal evolution of the variables
in their respective layers. These not statistically significant trends were explained by the high
variability and changes in the low-limit depth of the layers encountered between consecutive
years.
As there was a lack of in situ measurements and sampling along the west half of the Irminger
basin (36.5-42.5ºW) in the cruise of 2019 (due to permit restrictions to study the national
waters of Denmark), the GO-SHIP A25-OVIDE data for the cruise of 2018 (available at



SEANOE [https://www.seanoe.org/], Pascale et al., 2022) were considered to adjust the 2019
data. The average values were calculated with both the available data in the easternmost part
of the Irminger basin during the cruise of 2019 and the A25-OVIDE-2018 data available in
the same part of the section (29.6-36.5ºW). The difference between these average values
provides the variation of each variable from 2018 to 2019, which can be extrapolated to the
western part of the Irminger basin by assuming linearity in the temporal evolution. Thus, the
average values for 2019 were adjusted by applying the product with the calculated change
between 2018 and 2019.
**4. Discussion**
**4.1.    Reversal of the physical trends during the 2010s**
The present investigation revealed the cooling and freshening of the upper ocean in the
NASPG within the period 2009-2019 (Figure 4; Table 2), as recently reported since the reversal
of climatic trend and surface physical properties occurring after 2005 (Holliday et al., 2020;
Josey et al., 2018; Robson et al., 2016; Tesdal et al., 2018). The temperature decreased in the
upper ocean (with more than 95% level of confidence in SPMW, while non statistically
significant in ENACW) by 0.05-0.08 ºC yr$^{-1}$ (Table 2), which is consistent with the ratio of
heat loss per decade among the first 700 m depth equivalent to approximately -0.45 ºC decade$^{-1}$
$^{1}$ (-0.045 ºC yr$^{-1}$) encountered over the period 2005-2014 (Robson et al., 2016). The interannual
temperature trends in subsurface layers (Table 2) similarly draw the cooling observed in the
Irminger basin between 2008 and 2017 (-0.05 and -0.11 ºC yr$^{-1}$ for summer and winter,
respectively; Leseurre et al., 2020) and the winter average surface cooling along the entire
NASPG between 2004 and 2017 (-0.08 ± 0.02 ºC yr$^{-1}$; Fröb et al., 2019). The decrement in
subsurface salinity (with more than 95% level of confidence in both SPMW and ENACW) of
0.006-0.018 yr$^{-1}$ (Table 2) agreed with the interannual rates provided by Tesdal et al., (2018)
for the Irminger basin (-0.007 ± 0.002 yr$^{-1}$) and for the central-eastern NASPG (-0.020 ± 0.003
yr$^{-1}$) over the period 2004-2015.
The fluctuations in physical properties were linked to a decrease in oceanic heat transport and
storage within the NASPG, which has been attributed to changes in the AMOC over decadal
to multidecadal timescales (Balmaseda et al., 2007; Desbruyères et al., 2013; Mercier et al.,
2015; Smeed et al., 2018). However, the assessment of the temporal evolution of the AMOC





in high latitudes remains uncertain, and there is no evidence of its impact on physical patterns
across the NASPG on an interannual scale (Jackson et al., 2022). The changes in the
atmospherics forcing also account for the variability of the upper ocean physical properties
and can have a cumulative effect over several years (Balmaseda et al., 2007; Böning et al.,
2006; Eden and Willebrand, 2001; Marsh et al., 2005).
The distribution of the water mass properties, the processes of vertical and horizontal mixing
and the circulation patterns in the Irminger and Iceland basins were described by García-Ibáñez
et al., 2016 and 2018. The poleward path of the ENACW (Pollard et al., 1996) and its mixing
with waters moving from the west across the NASPG (Ellett et al., 1986) accounted for the
highest subsurface temperature and salinity signals observed in the Iceland basin and even
more in the Rockall. The SPMW and LSW in the Rockall Trough exhibited higher temperature
and salinity signals in the respectively order of ~1ºC and ~0.05-0.1 compared to the Irminger
and Iceland basins (Figure 4). The NASPG circulation patterns account for these differences
by transporting eastward these water masses, which subduct below the ENACW in the Rockall
Trough and mixed with warmer and more saline intermediate waters (i.e. Mediterranean
Water) moving from the south (e. g. Ellett et al., 1986; Harvey, 1982; Holliday et al., 2000).
The low temperature and salinity signals in the less-stratified Irminger basin (Figure 2)
experienced weaker interannual decreases in subsurface layers and higher rates of cooling and
freshening in intermediate and deep waters compared with the Iceland and Rockall basins
(Figure 4; Table 2). These longitudinal thermohaline heterogeneities were related to the
enhancement of vertical mixing processes in areas of water mass formation along the western
NASPG (Fröb et al., 2016; García-Ibáñez et al., 2015; Pickart et al., 2003; Piron et al., 2017)
and the water mass transformation along the NAC (Brambilla and Talley, 2008). The strongest
decrement in subsurface temperature and salinity along the Iceland and Rockall basins (Figure
4; Table 2) coincided with the significant event of heat loss and freshening observed by
Holliday et al., (2020) in the eastern NASPG over the period 2012-2016, so-called the Great
Salinity Anomaly. This pattern was not easily discernible in the Irminger basin due to the
transport of freshwater through the Fram Strait, as well as due the redirection of the Labrador
Current combined with changing wind stress curl (Holliday et al., 2020).



### 4.2. Evaluation of the interannual trends in $C_T$ in response to changes in $C_{ant}$ and $C_{nat}$

The changes in the physical patterns observed in the NASPG influenced the interannual variability of the MCS. The increase in $C_T$ expected in the upper ocean due to the atmospheric $CO_2$ uptake was offset by the cooling and freshening (and dealkalinization) of the subsurface layers in the entire NASPG. The entrance of $C_{ant}$ through the air-sea interface and its accumulation dominated the observed increase in $C_T$, while the $C_{nat}$ experienced a slightly decrease throughout the region (Figure 5 and Table 2). A detailed description of the interannual trends in $C_T$ and $A_T$ is provided in Appendix B.

The increase in the ventilation rates during this decade, shown by the negative AOU trends (Figure S6 and Table S4), explained the higher growth in $C_{ant}$ than expected due to the atmospheric $CO_2$ increase. It leads an enhancement in the vertical mixing processes which drove the transport of $C_T$-rich subsurface waters toward deeper areas and the slightly decrease in $C_{nat}$ through the whole water column. The trends of $C_{ant}$ among the 2010s (0.85-1.77 µmol kg$^{-1}$ yr$^{-1}$; statistically significant at the 99% level) were higher than the observed on a decadal to multidecadal scale since the late 20$^{th}$ century in the Irminger and Iceland basins (0.21-0.89 µmol kg$^{-1}$ yr$^{-1}$ during 1991-2015, García-Ibáñez et al., 2016; and 0.38-1.15 µmol kg$^{-1}$ yr$^{-1}$ during 1983-2013, Pérez et al., 2021), which suggest an enhancement in the $C_{ant}$ accumulation on interannual scales during periods of high ventilation, as previously reported by Perez et al., (2008).

The vertical distribution of $C_{ant}$ and $C_{nat}$ along the transect (Figure 3b and 3c) reflect the higher stratification in the Iceland and Rockall basin compared with the well vertically-mixed Irminger basin. It represents a source of variability in the interannual changes of $C_{ant}$ among the different layers and basins (Figure 4; Table 2). In the western NASPG, the surface heat loss and enhanced deep convection processes favour the solubility and subsequent uptake of atmospheric $CO_2$ and inject oxygenated and $CO_2$-rich waters into deeper layers (Messias et al., 2008). Its likely accounts for intermediate and deep layers in the Irminger basin exhibiting the highest $C_{ant}$ accumulation rates in the NASPG (Figure 5; Table 2). The highest ventilation of the interior ocean in the Irminger basin was demonstrated by its minimum AOU values (Figure 2 and S6). It induced a rapid surface-to-bottom transport of $C_{ant}$ shown by its highest rates of



increase in intermediate and deep waters throughout the region (Figure 5; Table 2). The high
$C_{ant}$ values and its rapidly increment at DSOW were explained by the improved oxygenation
of this layer at shallower depths (interannual AOU trends given in Table S4) and its subduction
through the continental slope below ISOW.
In the eastern NASPG, the stratification weakened due to the path of the NAC warming
eastward the upper water column and accounted to slowdown the increase in $C_{ant}$ in the Iceland
basin. An exception comes with the Rockall basin, in which the relatively warm and salty
ENACW (Figure 2 and 4) showed the maximum $C_{ant}$ (58-68 µmol kg$^{-1}$) and minimum $C_T$
(2120-2131 µmol kg$^{-1}$) and $C_{nat}$ (2058-2070 µmol kg$^{-1}$) throughout the region (Figure 3 and
5). The strong stratification of the Rockall Trough due to the wide differences in the physical
properties between the ENACW with SPMW and LSW plays a crucial role. The lower AOU
encountered in ENACW (<20 µmol kg$^{-1}$) compared with deeper layers (>30 µmol kg$^{-1}$) suggest
that the enhanced ventilation processes were limited to the subsurface layer increasing the
entrance of $C_{ant}$ through the air-sea interface. The strong oxygenation, which reach the oxygen
saturation after 2014, could be related with the high rates of renovation of ENACW due to its
path from the south (Pollard et al., 1996) and its mixing with waters moving eastward (Ellett
et al., 1986). As the NAC transports nutrient-rich waters northward and eastward into
subsurface layers in the Rockall Trough, biological production tends to increase and actively
reduced the $CO_2$ excess from the ENACW (McGrath et al., 2012b), as proved by the observed
low $C_T$ and $C_{nat}$. The strong interannual increase in the ENACW ventilation during this decade
increase the $C_{ant}$ and decrease the $C_{nat}$ (Rodgers et al., 2009) keeping approximately constant
the $C_T$ (Table 3). The poorly ventilated thermocline (AOU > 60 µmol kg$^{-1}$), placed between
500-1000 m in the eastern NASPG, induced a $C_{nat}$-driven increase in $C_T$ among the SPMW
and uLSW. However, its intrusion does not present relevant variations with time and thus does
not introduce differences in the interannual trends of the biogeochemical properties.

### 4.3.    Acidification trends

The interannual $pH_T$ trends (Figure 6, Table 2) exhibited the acidification of the whole water
column in NASPG during the period 2009-2019. Despite the acidification rates observed in
the most subsurface waters among the three basins were not significant at the 90% confidence
level (Table 2), they were consistent in the interval of 0.001 units yr$^{-1}$ to those observed during



larger periods at time-series stations located across the North Atlantic: at subtropical latitudes
($0.0018 \pm 0.0002$ units $yr^{-1}$ during 1995-2014 and $0.0020 \pm 0.0001$ units $yr^{-1}$ during 1995-2023
at ESTOC, González-Dávila and Santana-Casiano, 2023; and $0.0017 \pm 0.0001$ units $yr^{-1}$ during
1983-2014 at BATS, Bates et al., 2014) and subpolar latitudes ($-0.0017 \pm 0.0002$ units $yr^{-1}$ at
IRM-TS during 1983-2013 and $-0.0026 \pm 0.0002$ units $yr^{-1}$ at IS-TS during 1985-2013,
summarized by Pérez et al., 2021). In addition, the changes in the surface $pH_T$ trends has been
reported by Leseurre et al., (2020) in the western NASPG within a wide latitudinal area (54-
64ºN) during the period 2008-2017 in comparison with the periods 1993-1997 and 2001-2007.
Although the highly significant cooling observed in SPMW, the year-to-year variations in
ventilation (shown by the annual average AOU and its trends in Figure S6) and thus in $C_{nat}$ and
$C_{ant}$ (Figure 5), which could be related with fluctuations in the atmospheric forcing, introduced
relevant changes in $pH_T$ on an interannual scale and explained the low significant trends. This
behaviour was clearly reflected in the Irminger basin, where strong slowdowns in ventilation
were observed from 2009 to 2010 and from 2013 to 2014, resulted in a relatively increase in
$C_{nat}$ and decrease in $C_{ant}$ observed in SPMW and extended with less intensity through the whole
water column.
The highest acidification rates were found through intermediate and deep waters in the
Irminger and Iceland basins, coinciding with the highest rates of increase in $C_{ant}$ (Table 2,
trends statistically significant at more than 95% level of confidence). The exception comes
with the DSOW, which presented and interannual decrease in $pH_T$ in phase with those of the
uLSW. This singularity was previously observed by García-Ibáñez et al., (2016), which noticed
the similar trends between the DSOW and LSW attributed to the recently formation and sink
through the continental slope of the DSOW. The acidification rates found among the uLSW,
LSW and ISOW ($0.0026$-$0.0032$ units $yr^{-1}$) experienced, on an interannual scale, an
acceleration in comparison with previous reported based on long-term records [e. g. $0.0009$-
$0.0017$ units $yr^{-1}$ estimated for 1981-2008 by Vázquez-Rodríguez et al., (2012b); $0.0013$-
$0.0016$ units $yr^{-1}$ estimated for 1991-2015 by García-Ibáñez et al., (2016); $0.0015$-$0.0019$ units
$yr^{-1}$ estimated for 1983-2013 at the IRM-TS by Pérez et al., (2021); $0.0019 \pm 0.0001$ units $yr$-
1 estimated for 1993-2017 by Leseurre et al., (2020)]. Contrasting the rates of change in $pH_T$
during the decade of study with those encountered by these multidecadal evaluations (and
considering the total amount of years comprising each of the studies and the changes in the ion



hydrogen concentration- [$H_T^+$]), we estimate an acceleration in the rates of acidification of 0.4-
5.4% in the Irminger basin and 1.0-9.0% in the Iceland basin during the 2010s since the late
20th century. This acceleration was mainly attributed to increased deep-water ventilation
(shown in the rapid decrease in AOU in Figure S6) favouring the progressively increase in the
accumulation of $C_{ant}$ and $C_{nat}$ toward intermediate a deep layers, in which cooling was not
significant in the Irminger basin and neither enough intense in both basins to compensate the
acidification.
Although the similarities encountered in the $pH_T$ trends among both basins, the average values
presented differences which may be closely linked with the transport and transformations of
the water masses along the NASPG and mainly modulated by the Reykjanes Ridge (García-
Ibáñez et al., 2015, 2016, 2018). The transformation of the SPMW formed in the Iceland
(McCartney and Talley, 1982; Brambilla and Talley, 2008; Tsuchiya et al., 1992; Van Aken and
Becker, 1996) and flowing with the NAC across the Reykjanes Ridge (Brambilla and Talley,
2008) accounted for the lower $pH_T$ values in the Irminger basin. The differences in $pH_T$ found
at intermediate and deep layers were related with the divergence of the LSW path into two
cores when it reaches the Reykjanes Ridge (Álvarez et al., 2004; Pickart et al., 2003) and the
ISOW path flowing southward along the western Iceland basin and recirculated northward into
the eastern Irminger basin (Dickson and Brown, 1994; Saunders, 2001). These differences in
the spreading of water masses enhanced the ventilation in the Irminger basin favouring the fall
in $pH_T$ compared with the Iceland basin. The rise in the ISOW following the Reykjanes Ridge
slope through its eastern flank favoured a strong vertical mixing over and around the ridge
(Ferron et al., 2014) and a reduction of the LSW core in the Iceland basin (García-Ibáñez et
al., 2015), contributing to resemble $pH_T$ values and trends among the uLSW and LSW in this
basin.
The upper waters of the Rockall Trough presented the maximum $pH_T$ throughout the transect
(8.02-8.08 units). The observed strong $pH_T$ fluctuations between years related with interannual
changes in the NAC do not allow to discern trends with a statistically interval of confidence
equal or higher than the 90% on a decadal scale. The interannual decrease in $pH_T$ in the
ENACW (~0.001 units $yr^{-1}$) was half than the observed along southernmost transects in the
Rockall Trough between 1991 and 2010 (~0.002 units $yr^{-1}$, McGrath et al., 2012a). The



temporal distribution of the average $pH_T$ (Figure 6) highly influenced by changes in the
ventilation (seen in AOU trends in Figure S6) allow to discern two periods: the approximately
constant ventilation rates keep a steady state in terms of $pH_T$ during 2009-2011, while the
progressively renewal of oxygenated water after 2012 (and peaking in this year) increase the
$pH_T$. The year-to-year variability in the biogeochemical patterns after 2012 may be attributed
to the fluctuations in the spreading into the Rockall Trough of several water masses occupying
different depths coming from the south and east (Ellett et al., 1986; Pollard et al., 1996). This
contributed to enhance the oxygenation of the ENACW during the 2010s (seen in minimum
AOU values highly variables between years and which tend to decrease with 99% statistical
confidence; Figure S6 and Table S4) and the reduction of the injection of saline subsurface
waters from subtropical latitudes (Holiday et al. 2020). The findings suggest that the strong
decrease in $A_T$ (Figure S4 and Table S4) due to the freshening and weak increase in $C_T$ (Figure
5 and Table 2) due to enhanced ventilation counteract the acidification in the ENACW. The
SPMW among the Iceland and Rockall basins showed similar $pH_T$ trends (Table 2) due to the
emplacement of the poorly-oxygenated thermocline at these depths (García-Ibáñez et al.,
2016). The approximately constant AOU at SPMW in the eastern NASPG (Figure S6) proved
its steady ventilation, which can introduce differences in the acidification rates among the
layers accomplishing the Rockall Trough. The influence of the cooling and freshening of
deeper areas due to the spreading and horizontal mixing was notable in the LSW, which
presented slightly higher $pH_T$ values in the Rockall respect to the adjacent Iceland basin.

### 4.4.  Drivers pH

Due to the variety of processes involved in OA, a decomposition of the $pH_T$ trends into the
individual components that govern its spatio-temporal variability was done (see section 2.2.5).
The interannual $pH_T$ changes $\left(\frac{dpH_T}{dt}\right)$ explained by fluctuations in temperature $\left(\frac{\partial pH_T}{\partial T}\frac{\partial T}{dt}\right)$, salinity
$\left(\frac{\partial pH_T}{\partial S}\frac{\partial S}{dt}\right)$, $A_T$ $\left(\frac{\partial pH_T}{\partial A_T}\frac{\partial A_T}{dt}\right)$ and $C_T$ $\left(\frac{\partial pH_T}{\partial C_T}\frac{\partial C_T}{dt}\right)$ were calculated for each layer and basin (Eq. 2) and
summarized in Table 3. The positive contributions of each of the drivers indicate an increase
in $pH_T$ while negative contributions the opposite. The cumulative $pH_T$ change resulting from
the distinct drivers $\left(\frac{dpH_T}{dt}\right.$ (calculated) in Table 3) were consistent with the observed $pH_T$ trends
$\left(\frac{dpH_T}{dt}\right.$ (obs) in Table 3, discussed in section 4.2), thereby instilling confidence in the
methodology. The minimal differences between observed and calculated rates of change have





added coherence to the non-significant trends identified for pH and its drivers in some basins
and layers (Table 2, 3 and S4). In the entire section at SPMW, the $\frac{dpH_T}{dt}$ (calculated), explained
by the cumulative impact of its drivers (all of them statistically significant at the 95% level of
confidence), aligns within a range of <0.0002 units yr$^{-1}$ with $\frac{dpH_T}{dt}$ (obs) (which was not
significant). In the Irminger and Iceland basins at intermediate and deep layers, the $\frac{dpH_T}{dt}$ (obs)
(statistically significant at least at the 95% level of confidence) were consistent within the
range of <0.001 units yr$^{-1}$ with $\frac{dpH_T}{dt}$ (calculated) (T, S and NA$_T$ shows non-significant trends at
some of the intermediate and deep layers). The interannual variations were non-significant for
pH$_T$ neither for its drivers in the Rockall Trough at LSW and ENACW. The high temporal
dispersion of average data in these layers was mainly related to the rise in depth of LSW along
the eastern continental slope and its mixing with shallower waters coming from subtropical
latitudes (Ellett et al., 1986; Harvey, 1982; Holliday et al., 2000). The substantial variability
introduced by these processes made it difficult to discern the pattern of acidification and its
drivers on an interannual scale in the shallow Rockall Trough. Therefore, long-term monitoring
and the development of multidecadal-scale studies are required in this area to derive significant
conclusions.
The cooling and freshening of the NASPG during the 2010s modified the physical-driven pH$_T$
changes compared with those encountered by García-Ibáñez et al., (2016) during previous
decades in the western NASPG. The cooling contributed to increase the pH$_T$ and compensated
the observed acidification rate. The increase in pH$_T$ due to temperature fluctuations was
maximum at SPMW (~0.001 units yr$^{-1}$) and was reduced an order of magnitude to negligible
toward intermediate and deep layers (<0.0003 units yr$^{-1}$ at uLSW and below). The increase in
pH$_T$ due to salinity fluctuations was minimal (<0.0001 units yr$^{-1}$) through the whole water
column in the three basins, reflecting that the observed freshening caused insignificant changes
in pH$_T$. The temperature and salinity contributed by 19.1-26.5% and 1.2-3.3%, respectively, in
the total pH$_T$ change in the upper layers, while presented an influence three times lower in
intermediate and deep layers (1.3-7.6% and <0.6%, respectively). The enhanced convective
processes in the Irminger basin (e. g. Fröb et al., 2016; García-Ibáñez et al., 2015; Gladyshev
et al., 2016a, 2016b; Piron et al., 2017) together with the rapid transport of LSW from the
Labrador Sea to the Irminger basin (Yashayaev et al., 2007) introduced differences in the



thermal-driven pH$_T$ with the Iceland basin which has been previously reported by García-
Ibáñez et al., (2016). The advection of LSW through the Greenland continental slope also
affected the DSOW (Read, 2000; Yashayaev and Dickson, 2008), which shows thermal-driven
pH$_T$ changes consistent with those encountered through the LSW in the Irminger basin.
Despite the negligible direct contribution of the salinity fluctuations over the pH$_T$ changes, the
freshwaters fluxes influence the distribution of $A_T$ and $C_T$ indirectly affecting pH$_T$ trends. Once
removed the effect of salinity by normalization (Pérez et al., 2021), the positive $NA_T$ trends
encountered in the upper layers lead a rise in pH$_T$, while the diminished $NA_T$ contributed to
decrease the pH$_T$ toward the interior ocean. The changes in $NA_T$ described the 7.8-10.1% of
the total pH$_T$ change at SPMW. The $NA_T$-driven pH$_T$ changes became insignificant with depth
(Table 3) due to the insignificantly interannual changes in $NA_T$ through LSW and ISOW (Table
S4). The weak contribution of $NA_T$ in these layers (1.3-5.1%) could be related to the difficulty
of reversing the large alkalinization until the 2000s resulted from the slowdown in the
formation of LSW since the mid-90s (Lazier et al., 2002; Yashayaev, 2007), which was
transmitted towards deeper overflow waters (Sarafanov et al., 2010). The substantial
interannual changes and the abrupt change between periods of increase and decrease of the
seawater properties at DSOW (Yashayaev et al., 2003; Stramma et al., 2004) linked with
changes in the LSW formation (Dickson et al., 2002) explained the rapidly decrease in $NA_T$
(Table S4), which described the 14.6% of the pH$_T$ declining.
The increase in $NC_T$ drove by the rise in $C_{ant}$ was found to govern the acidification, with a
contribution higher than the 67% in the whole water column throughout the region. The $NC_T$-
driven pH$_T$ declining was close to twice the observed and calculated acidification rates through
the SPMW (Table 3). However, the contribution of $NC_T$ at SPMW (67-69%) was lower than
the encountered toward the interior ocean (82-96%) due to the relevance of temperature and
$A_T$ over pH$_T$ trends in the upper ocean. The cooling and increase in $NA_T$ counteracted the
acidification expected by the increasing $C_T$ at SPMW by 28-34% and 11-15%, respectively.
The weaker cooling through the intermediate and deep layers leads a lower thermal-
neutralization of the $C_T$-driven acidification (1.5-9.3%), while the decreasing $NA_T$ contributed
to decrease the pH$_T$ by < 2-12% in the uLSW, LSW and ISOW and by ~15% in the DSOW.
The driver analysis also remarked that the role of freshening in counteract the acidification



was small in the upper layers (<6%) and becoming insignificant toward the interior ocean
(<2%).

### 4.5.    Interannual changes in $\Omega_{Ca}$ and $\Omega_{Arag}$

The analysis of the changes in $\Omega_{Ca}$ and $\Omega_{Arag}$ hold significance in elucidating the potential
effects of OA over the $CaCO_3$ species calcite and aragonite, thereby offering insights into their
potential implications for marine calcifying organisms and ecosystems. The vertical
distribution of $\Omega_{Ca}$ and $\Omega_{Arag}$ is presented in Figure S3. The upper and intermediate layers up
to 2100-2400 m depth of the Irminger and Iceland and the whole Rockall basin were
supersaturated for aragonite ($\Omega_{Arag}$ >1), while the DSOW was undersaturated ($\Omega_{Arag}$<1). The
ISOW, with $\Omega_{Arag}$ ranged between 1.0 and 1.1 at the beginning of the decade, crossed to
undersaturated conditions at the end of the period due to the progressively rise of the aragonite
saturation horizon (depth in which $\Omega_{Arag}$=1). The whole water column throughout the section
was supersaturated for calcite ($\Omega_{Ca}$>1) due to its lower solubility (Mucci, 1983). The $\Omega_{Ca}$ and
$\Omega_{Arag}$ in the SPMW (2.2-2.7 and 1.4-1.7 units, respectively) were lower than the encountered
equatorward in the subsurface Atlantic (>4.0 and >2.5 units, respectively; González-Dávila et
al., 2010; González-Dávila and Santana-Casiano, 2023). The poleward pathway of low-
latitude upper waters through the Rockall Trough explained the higher $\Omega_{Ca}$ and $\Omega_{Arag}$ found in
the ENACW (3.0-3.6 and 1.8-2.3 units, respectively). The reduction in $\Omega_{Ca}$ and $\Omega_{Arag}$ towards
higher latitudes in upper and intermediate layers smooth the vertical gradients in the NASPG
compared with the subtropical latitudes (González-Dávila et al., 2010; González-Dávila and
Santana-Casiano, 2023).
The interannual trends in $\Omega_{Ca}$ and $\Omega_{Arag}$ (Figure 7, Table 2) exhibited the decrement through
the whole water column along the NASPG with a level of statistical confidence generally
higher than the 90%. The rates of declining for $\Omega_{Ca}$ and $\Omega_{Arag}$ in the SPMW (0.011-0.021 and
0.007-0.013 units yr-1; respectively) were consistent with the trends observed up to 100 m
depth at ESTOC between 1995 and 2023 (0.019 ± 0.001 and 0.012 ± 0.001   units yr$^{-1}$,
respectively; González-Dávila and Santana-Casiano, 2023) and in surface waters at the IS-TS
between 1985 and 2008 (0.0117 ± 0.0011 and 0.0072 ± 0.0007 units yr$^{-1}$, respectively;
Olafsson et al., 2009). The declining in $\Omega_{Arag}$ in the SPMW accelerated by ~26% and ~51% in
the Irminger and Iceland basins, respectively, in comparison with the trends given for the



period 1991-2018 ($0.0052 \pm 0.0006$ and $0.0049 \pm 0.0015$ units yr$^{-1}$, respectively; García-Ibáñez
et al., 2021). The observed decrease in $\Omega_{Arag}$ in the SPMW was ~23% faster in the Rockall
Trough than in the adjacent Iceland basin. The interannual declining for $\Omega_{Ca}$ and $\Omega_{Arag}$ in the
ENACW (0.012 and 0.008 units yr$^{-1}$, respectively) agreed with these previous observations but
were not statistically significances likely due to the high variability modifying the changes in
$pH_T$ in this layer (see section 4.2). Despite the acceleration of the acidification rates toward
intermediate and deep layers, the declining rates weakened for $\Omega_{Ca}$ and even more for $\Omega_{Arag}$
(Table 2). Moreover, the vertical profiles were approximately constant throughout the section
in contrast with the heterogeneous vertical distribution of $pH_T$ between basins. This behaviour
was previously observed in the Irminger and Iceland basins by García-Ibáñez et al., (2021) and
explained by pressure and temperature-induced changes in the speciation of the $CO_2$-carbonate
chemistry species (Jiang et al., 2015) and in the solubility of calcite and aragonite (Mucci,
1983). Their combined action counterbalanced the alterations in $\Omega$ resulting from acidification,
particularly in colder deep waters where the solubility of calcite and aragonite was reduced
(García-Ibáñez et al., 2021). However, the fall down in $\Omega_{Ca}$ and $\Omega_{Arag}$ along the uLSW, LSW
and ISOW accelerated by 40-75% in relation with the trends reported by García-Ibáñez et al.,
(2021) for the Irminger and Iceland basins. The LSW and ISOW presented faster declining
rates for $\Omega_{Ca}$ and $\Omega_{Arag}$ in the Irminger (Table 2), which may be caused by the enhanced
ventilation of the interior ocean which accelerated the acidification (see section 4.2). The
westward rise in depth of these layers along the Greenland continental slope, accompanied by
a subsequent elevation in the horizons of solubility, resulted in reduced buffering capacity
against acidification effects in the Irminger basin when compared to the Iceland basin. In
contrast, the rise in depth of LSW in the Rockall Trough favour the increment of ~0.2 units in
$\Omega_{Ca}$ and $\Omega$Arag with respect to the Iceland basin but had not influence on the interannual
trends, which were coinciding. The $\Omega_{Ca}$ and $\Omega_{Arag}$ in the DSOW, despite showed a trend
accelerated by ~30% compared to the observed by García-Ibáñez et al., (2021), presented the
weakest interannual decreases throughout the section ($0.004 \pm 0.003$ and $0.002 \pm 0.001$ units
yr$^{-1}$, respectively) due to the high pressure and low temperatures compensating the rapidly
acidification (Figure 6, Table 2).
A driver analysis enabled the assessment of the impact of individual processes involved in OA
on the variations in $\Omega_{Ca}$ and $\Omega_{Arag}$ (see section 2.2.6). The correlation of $\Omega$ with $pH_T$ ($r^2=0.90$)



with a level of significance higher than the 99% explained that the individual components
driving OA accompanied the declining in $\Omega$. The interannual $\Omega$ variations ($\frac{d'\Omega}{dt}$) explained by
fluctuations in temperature ($\frac{\partial'\Omega}{\partial T}\frac{\partial T}{dt}$), salinity ($\frac{\partial'\Omega}{\partial S}\frac{\partial S}{dt}$), $A_T$ ($\frac{\partial'\Omega}{\partial A_T}\frac{\partial NA_T}{dt}$) and $C_T$ ($\frac{\partial'\Omega}{\partial C_T}\frac{\partial NC_T}{dt}$) were calculated
for each layer and basin (Eq. 5) and summarized in Table 4. The sum of changes in $\Omega$ due to
the distinct drivers ($\frac{d'\Omega}{dt}$ (calculated) in Table 4) agreed with observed $\Omega$ trends ($\frac{d'\Omega}{dt}$ (obs) in Table
4) in all the basin and layers except for the DSOW, in which the strong $NA_T$ decrease had a
crucial influence on declining $\Omega$. The driver analysis, as mentioned when was applied for pH$_T$,
contributes to add coherence and consistency to those non-significant trends identified and/or
its drivers in some basins and layers (Table 2, 3 and S4)
The $C_{ant}$-driven rise in $NC_T$ governed the decrease in $\Omega$ with a contribution of 79-83% in the
SPMW which reached ~97% toward deeper waters. The increase in $NA_T$ in the SPMW
accounted by 10.4-13.0% in the $\Omega$ trends and counteracted its $NC_T$-driven decrease by 12.6-
16.2%. The contribution of $NA_T$ fall and reversed toward deeper waters, explained <6% of the
decline in $\Omega$ in the uLSW, LSW and ISOW in the Irminger basin and <11% in the Iceland
basin. The pronounced impact of the rapid decrease in $NA_T$ on the acidification of the DSOW
(see section 4.3) depicted the greater contribution of $NA_T$ encountered among the Irminger
basin (16%) and compensated the $C_T$-driven decrease in $\Omega$ by 36.4%. In the Rockall Trough,
the contribution of $NC_T$ changes on $\Omega$ was reduced at LSW (78.2-79.0%) compared to the
Irminger basin (94.5%) while the effect of $NA_T$ fluctuations tripled until reach 12.6-12.7%.
Despite the evaluated crucial role of cooling in counteracting the acidification, the temperature
fluctuations have an opposite effect on $\Omega$ owing to the thermodynamic relationship inherent in
the acid-base equilibrium of the $CO_2$-carbonate system (Dickson and Millero, 1987). In the
Irminger and Iceland basins, the observed decreasing temperatures negligibly contributed to
fall down the $\Omega$ (3.6% in the SPMW and <2% in intermediate and deep waters). The influence
of salinity, as occurred with the pH$_T$ trends, was minimal: the observed freshening contributed
to elevate the $\Omega$ trends and compensated its declining by 4.6-4.7% at SPMW, 1.1-2.1% at
uLSW and LSW and 0.5-1.2% at ISOW and DSOW. Even the slightly faster cooling and
freshening observed in the Rockall Trough, the contributions of temperature and salinity on
the $\Omega$ did not exceed the 7% in each of its layers.



The driver analysis exhibited the strongest interannual decrease in $\Omega$ in the upper layers governed by the uptake of $C_{ant}$ weakly compensated by the increase in $NA_T$ and favoured by the cooling and freshening. The decrease in $\Omega$ could have severe consequences on organisms reliant on aragonite, which is less resistant to dissolution than calcite (Mucci, 1983; Broecker and Peng, 1983) and thus expected to experience relatively higher susceptibility to the effects of OA over shorter time scales (Raven et al., 2005). The progressive reduction in $\Omega_{Arag}$ is driving a long-term decrease in the depth of the aragonite saturation horizon ($\Omega_{Arag}=1$) by 80-400 m since the preindustrial era (Álvarez et al., 2003; Feely et al., 2004; Pérez et al.,2013, 2018; Pérez et al., 2013; Tanhua et al., 2007; Wallace, 2001) and is projected to shoal by more than 2000 m by the end of the century under the IS92a scenario (Orr et al., 2005). Likewise, Orr et al., (2005) suggested that high-latitudes surface waters could become undersaturated when the atmospheric $CO_2$ concentration double the preindustrial concentration within the next 50 years. It would reduce the calcification rates in some shallow calcifying organism by more than the 50% (Feely et al., 2004).

The planktonic aragonite-producers pteropods (e. g. *Limacina helicina, Clio pyramidata*), which have high population densities in subpolar regions up 300 m depth (Bathmann et al., 1991; Urban-Rich et al., 2001) and play a key role in the export flux of both carbonate and organic carbon (Accornero et al., 2003; Collier et al., 2000), are expected to be highly vulnerable to OA if the aragonite saturation horizon continue to shoal (Orr et al., 2005). The undersaturation toward intermediate and upper layers negatively influence the aragonite-based CWC (e. g. *Lophelia pertusa, Madrepora oculate*), which show their highest diversity and population along the NASPG between 200 and 1000 m depth among the global ocean (Roberts et al., 2009). In fact, several studies reported that CWC ecosystems are anticipated to be among the first deep-sea ecosystems to experience acidification threats (Guinotte et al., 2006; Maier et al., 2009; Raven et al., 2005; Roberts et al., 2009; Turley et al., 2007), particularly in the North Atlantic (Perez et al., 2018). The findings presented here contribute to a deeper understanding of the biological impacts of OA along the NASPG.

**5. Conclusions**

This research has evaluated the interannual changes in the basin-wide $CO_2$-carbonate system dynamics along the NASPG during the 2010s. Despite the observational period is relatively





short to quantify long-term trends and to formulate significant future projections, the finding
has allowed to evaluate the ocean response, in terms of carbonate system dynamics and on an
interannual scale, to changes in deep-water convection and to isolate events affecting the
physical patterns. The present study improved the comprehension of how the processes
modifying the rates of accumulation of $C_{ant}$ and acidification on an interannual scale could
have a relevance impact on its decadal and multidecadal trends.
The assessment of OA within the Irminger and Iceland basins was enhanced by supplying
novel data and trends spanning a decade in which the physical patterns reversed. Additionally,
the study provides an unprecedent analysis of the physico-chemical variations in the Rockall
Trough, which is crucial for the assessment of the entire longitudinal span of the NASPG and
advancing our understanding of OA in the North Atlantic and Global Ocean. The data and
results given in this article could be used for modelling and compared with other repeated
hydrographic section data at mid and high latitudes in the North Atlantic, such as the A02,
A25, AR07E and AR28 framed in the GO-SHIP program, as well as used in conjunction to
develop future studies focused on the transport of $C_{ant}$-loaded and acidified waters. The
observational period is relatively short to quantify long-term trends and to formulate
significant future projections. The acceleration in surface warming and consequent changes in
$f\mathrm{CO_2}$ and pH observed during 2010s may be linked to isolated extreme events such as marine
heat waves and are not necessarily indicative of prolonged behaviours over time.
Overall, the entrance and accumulation of $C_{ant}$ and interannual acidification trends were
strongly affected by the cooling, freshening and enhancement in the oxygenation of the whole
water column during the 2010s. The interannual acidification trends ranged between 0.0013
and 0.0032 units $yr^{-1}$ in the Irminger basin, 0.0023 and 0.0029 units $yr^{-1}$ in the Iceland basin
and 0.0006 and 0.0024 units $yr^{-1}$ in the Rockall Trough. The convective processes increased
the accumulation rates of $C_{ant}$ in the interior ocean by 50-86% and accelerated the acidification
rates by around 10% compared to previous decades in the Irminger and Iceland basins. In the
eastern NASPG, the shallower hydrography of the Rockall Trough and the poleward
circulation patterns accounted for differences in the acidification rates respect to surrounding
waters. The high variability of this area explained the non-significant trends at interannual
timescales and support the necessity of assess the evolution of its carbonate system properties



over larger time periods. However, the low $NA_T$ content of ENACW due to the spreading of
subtropical subsurface waters into higher latitudes was suggested as the main process
decelerating the acidification trends in the upper Rockall Trough. The improved oxygenation
of LSW decreasing the $C_{nat}$ and thus compensating the $C_{ant}$-driven increase in $C_T$ may
contributed to slowdown the declining in $pH_T$ in relation to the Iceland basin. The acidification
of the NASPG was accompanied by a decline in the $\Omega_{Ca}$ and $\Omega_{Arag}$ of 0.004-0.011 and 0.003-
0.009 units yr$^{-1}$, respectively, in the Irminger basin; 0.007-0.016 and 0.005-0.010 units yr$^{-1}$,
respectively, in the Iceland basin; and 0.008-0.021 and 0.005-0.013 units yr$^{-1}$, respectively, in
the Rockall Trough.
The rise in $NC_T$, mainly explained by the increasing uptake of $C_{ant}$, was found to govern the
acidification of the NASPG with a contribution ranged between 53% and 68% in the upper
water column and higher than 82% toward the interior ocean. The increase in $NC_T$ was also
the main driver of $\Omega_{Ca}$ and $\Omega_{Arag}$ trends, with contributions higher than 82% in the Irminger
basin, 79% in the Iceland basin and 64% in the Rockall Trough. The combined effect of the
decreasing temperature, salinity and $NA_T$ neutralized the 45-49% of the $C_T$-driven acidification
along the entire longitudinal span of the SPMW. The cooling drove this compensation (27-
50%) followed by the decrease in $NA_T$ (11-33%), while the freshening had a minimal influence
(<6%). The deep-water ventilation processes slowdown the cooling and freshening toward the
interior ocean in the Irminger and Iceland and drove the progressively interannual increase in
$NA_T$. Thus, the $NA_T$ contributed to acidification by <11% within the intermediate and deep
layers and the physical counteraction of the $C_T$-driven acidification fall to <10%. In contrast,
the cooling weakly promoted the decline in $\Omega$ (<7% in the upper water column and <2% toward
the interior ocean), being only efficiently counteracted in subsurface layers by the increase in
$NA_T$ (12-16%) and the freshening (3-5%).
The present investigation pretended to emphasize the progressively increase in the uptake and
accumulation of $C_{ant}$ and subsequent acceleration of OA along the NASPG. The longitudinal
span of the NASPG and the differences in circulation patterns, water masses and bathymetry
along the section behave as a relevant source of spatio-temporal variability. The enhanced
convective processes in the western NASPG were found to favour the entrance of $C_{ant}$ in
intermediate and deep-layers and this its acidification, as well as influence the carbonate





system dynamics in the eastern NASPG. The advancement of comprehensive basin-wide
longitudinal evaluations, as the presented here, facilitates a more accurate understanding of the
mechanisms dictating basin-scale acidification processes. Furthermore, this promotes the
improvement of the projections pertaining to the future state of the oceans run by models and
forecast. Considering the important variability in the mechanism controlling the distribution
of the physico-biogeochemical properties and particularly the OA in the North Atlantic, this
research aims to highlight the necessity of continue monitoring and sampling the whole water
column through repeated hydrographic sections, especially through the highly variable but less
assess easternmost part.

**Appendix A: Correction of Dissolved Oxygen records for the cruise of 2019**

The sensor-measured DO data for the cruise of 2019 were corrected by considering the DO
output data given by the neural network ESPER_NN (Carter et al., 2021) for the cruises of
2016 and 2019 (hereinafter ESPER-estimated DO) and the WINKLER-measured DO during
the cruise of 2016. Among the 16 equations provided by the ESPER_NN that differently
combines seawater properties as predictors, we use the equation 8 which only need as inputs
the T and S (due to lack of measured macronutrients during the cruise of 2019) along with
latitude, longitude, depth and date (see Table 2 in Carter et al., 2021). The reported Root Mean
Squared Error (RMSE) of equation 8 for DO estimations in the global ocean is $\pm$ 9.7 µmol kg-
1, which is reduced for intermediate waters (1000-1500 m) to $\pm$ 5.9 µmol kg$^{-1}$ (see Table 7 in
Carter et al., 2021). Additionally, a new set of DO for 2019 based on WINKLER data for 2016
was computed, which was referred in this study as "pseudo-WINKLER" data. The difference
between WINKLER-measured and ESPER-estimated DO during 2016 was interpolated to the
longitudes and depths of the samples of 2019 by applying Delaunay Triangulation. The
pseudo-WINKLER data was described as the sum of these interpolated differences and the
ESPER-estimated DO data for 2019. The longitudinal distribution of measured and ESPER-
estimated DO data for 2016 and 2019 is depicted in Figure S1a and S1b. The interpolated
pseudo-WINKLER data for the cruise of 2019 were included in Figure S1a.
The sensor records of DO in 2019 were in average 4.90 µmol kg$^{-1}$ lower than the ESPER-
estimated and 10.31 µmol kg$^{-1}$ lower than the pseudo-WINKLER. A higher discrepancy was
observed in the average sensor-measured DO in the east part (237.60 $\pm$ 15.00 µmol kg$^{-1}$)

 

compared with the west part (281.40 ± 14.75 µmol kg⁻¹). The average differences (measured
minus ESPER-estimated DO and measured minus pseudo-WINKLER DO, $\Delta DO_{\text{meas-ESPER}}$ and
$\Delta DO_{\text{meas-pseudoWINLKER}}$, respectively; Figure S2c and S1d) shows that the sensor records were
strongly underestimated in the east part (-20.98 ± 10.91 and -28.77 ± 12.60 µmol kg⁻¹,
respectively) and weakly overestimated in the west part (8.59 ± 8.53 and 5.18 ± 12.02 µmol
kg⁻¹, respectively) during the cruise of 2019. These differences were corrected separately west
and east of 21.5ºW by using the relationship $\frac{\Delta DO_{\text{meas-pseudoWINKLER}}}{\text{measured DO}}$. The averages of this
relationship in the west and east part of the transect (0.016 and -0.12 µmol kg⁻¹, respectively)
were used as corrector factors. The corrected DO values were given by the product between
the measured DO and $\left(1 - \frac{\Delta DO_{\text{meas-pseudoWINKLER}}}{\text{measured DO}}\right)$.
**Appendix B: Interannual trends of $C_T$, $NC_T$, $A_T$ and $NA_T$**
The observed rates of increase in $C_T$ (Table 2) did not show notable differences with respect
to the interannual trends determined from previous decades at the Irminger and Iceland basins
(0.62-0.82 and 0.38-0.64 µmol kg⁻¹ yr⁻¹, respectively; García-Ibáñez et al., 2016) and at IRM-
TS and IS-TS (0.49-0.71 and 0.39-0.94 µmol kg⁻¹ yr⁻¹, respectively; Pérez et al., 2021). The
interannual rates of increase in $NC_T$ were higher than those of $C_T$ in the subsurface layers,
while the trends were similar among intermediate and deep layers (Table 2).
The interannual trends of $A_T$ (Figure S4 and Table S4) was found to be highly impacted by
freshening, with decreasing rates ranging from -0.33 to -0.71 µmol kg⁻¹ yr⁻¹ among the
SPMW and ENACW and from -0.01 to -0.18 µmol kg⁻¹ yr⁻¹ within the uLSW, LSW, ISOW
and DSOW. It contrasts with the minimal interannual changes and slight rates of increase in
$A_T$ encountered among the different layers by García-Ibáñez et al., (2016) from 1991 to 2015
in the Irminger basin (between 0.10 and 0.28 µmol kg⁻¹ yr⁻¹) and Iceland basin (between -
0.04 and 0.07 µmol kg⁻¹ yr⁻¹), and with the trends reported for the period 1983-2013 by Pérez
et al., (2021) at the IRM-TS (between 0.13 and 0.22 µmol kg⁻¹ yr⁻¹) and at the IS-TS (between
-0.04 and 0.15 µmol kg⁻¹ yr⁻¹). These heterogeneities in the temporal evolution of the $A_T$ were
driven by the decadal salinification of the whole water column observed since the late 20th
century and interrupted by interannual freshening episodes such as during the 2010s.



The interannual $NA_T$ trends reversed in comparison with those of $A_T$ along the SPMW and
ENACW (Figure S4 and Table S4). This increment in $NA_T$ was related with the stagnation of
$A_T$-rich subtropical waters in the upper layers due to the slowdown of the NASPG since the
mid-90s (e. g. Böning et al., 2006; Häkkinen and Rhines, 2004).
**Code Availability**
MATLAB and R codes for CANYON-B are available at
https://github.com/HCBScienceProducts/CANYON-B. MATLAB and R code for ESPER_NN
are available at https://github.com/BRCScienceProducts/ESPER. MATLAB code for
anthropogenic carbon calculation is available at
http://oceano.iim.csic.es/_media/cantphict0_toolbox_20190213.zip. The $CO_2SYS$ programme
for MATLAB is available at https://github.com/jonathansharp/CO2-System-Extd.
**Data Availability Statement**
The measured surface-to-bottom CLIVAR data (2009-2019) used in this investigation are
published in open-access at Zenodo (DOI: 10.5281/zenodo.10276221). The GO-SHIP A25-
OVIDE data for the cruise of 2018 is available at SEANOE
(https://www.seanoe.org/data/00762/87394/).
**Author contribution**
DCH contributed with data analysis and wrote the manuscript. FFP, DCH, AV, DGS, AGG,
MGD and JMSC worked on the design, conceptualization and data preparation. SG, AS, MGD,
JMSC, AGG and DGS participated in 8, 4, 7, 7, 2 and 2 cruises, respectively. SG and AS were
the Chief Scientist in all cruises and responsible for the operational and maintenance
procedures for the CTD and additional sensors and thus for physical and sensor-measured
variables. MGD and JMSC got the funding acquisition and provision of resources for the
Spanish team from the ULPGC. SG and AS got the funding for ship time and provision of
resources for all the cruise participants. All authors critically revised the manuscript.
**Competing interest**
The authors declare that the research was conducted in the absence of any commercial or
financial relationships that could be construed as a potential conflict of interest.
**Acknowledgement**
The participation on the cruises for the Spanish Team from the ULPGC was funded by the
Science Spanish Ministry under the Complimentary Actions CTM2008-05255, CTM2010-
09514-E and CTM2011-12984-E (years 2009-2011), the FP7 European project



CARBOCHANGE under grant agreement no. 264879 and by the Spanish Innovation and
Science Ministry through the Projects EACFe (CTM2014-52342-P) and ATOPFe (CTM2017-
83476-P). SG and AS were supported by FMWE-2023-0002. FFP and AV were supported by
the      BOCATS2      (PID2019-104279GB-C21)      project      funded      by
MCIN/AEI/10.13039/501100011033 and by EuroGO-SHIP project (Horizon Europe
#101094690). The participation of DCH was funded by the PhD grant PIFULPGC-2020-2
ARTHUM-2. Special thanks go to the technician and researchers Adrian Castro Álamo (2
cruises), Anna Barrera Galderique (3 cruises), Rayco Alvarado Medina (2 cruises) and Pilar
Aparicio Rizzo (1 cruise) who helped with in situ analysis. We also thanks technicians at the
P. P. Shirshov Institute of Oceanology from the Russian Academy of Science for the in situ
analysis of dissolved oxygen and nutrients.



**Legend for figures**

Figure 1. (a) Map of the North Atlantic Subpolar Gyre (NASPG) with the schematic diagram of the surface and deep circulation patterns compiled from Lherminier et al., (2010); Pérez et al., (2021); Sarafanov et al., (2012); Schmitz and McCartney, (1993); Schott and Brandt, (2007) and Sutherland and Pickart, (2008). The acronyms are defined as follow: the bathymetric features are shown in grey (RR: Reykjanes Ridge, HB: Haton Bank, GBB: George Bligh Bank, CGFZ: Charlie-Gibbs Fracture Zone, GIR: Greenland-Iceland Ridge, and GSR: Greenland-Scotland Ridge), the surface currents are shown in orange (NAC: North Atlantic Current, and IC: Irminger Current) and the deep water circulation is shown in blue and purple (ISOW: Iceland-Scotland Overflow Water, DSOW: Denmark Strait Overflow Water, LSW: Labrador Sea Water, and DWBC: Deep Western Boundary Current). The longitudinal distribution of the surface-to-bottom sampling stations along the cruise track of 2016 (repeated throughout the cruises) is shown with red dots. The black lines along the cruise track delimited the three basins. (b) Vertical distribution of the water masses considered in this study for each of the basins. The isopycnals, plotted over the salinity distribution for the cruise of 2016, show the limits of the layers and were defined by potential density (in kg m$^{-3}$) referred to 0 dbar ($\sigma_0$). The vertical gray lines show the limits between basins. The acronyms of the water masses and the selection of potential density values delimiting the layers are detailed in section 2.2.4. Figure produced with Ocean Data View (Schlitzer, 2021).

Figure 2. Water-column distribution along the longitudinal transect of (a) temperature, (b) salinity, (c) $A_T$, (d) $pH_T$ andI) AOU for the cruises of 2009 (left plots) and 2016 (right plots). The vertical while lines show the limits between basins. Figure produced with Ocean Data View (Schlitzer, 2021).

Figure 3. Water-column distribution along the longitudinal transect of (a) $C_T$, (b) $C_{ant}$ and (c) $C_{nat}$ for the cruises of 2009 (left plots) and 2016 (right plots). The vertical while lines show the limits between basins. Figure produced with Ocean Data View (Schlitzer, 2021).

Figure 4. Temporal distribution (2009-2019) of the average temperature and salinity in each of the layers considered for the Irminger (left plot column), Iceland (central plot column) and Rockall basins (right plot column). The average values were calculated for each cruise and layer and represented with coloured points together with their respective error bars at the time of each cruise (the method used for calculations was described in section 3.2). In the Irminger plots, the empty points represent the average values for 2019 calculated with the measured data available in the easternmost part of the basin (sampled part during this cruise), while the coloured points for 2019 represent the average values corrected with A25-OVIDE-2018 data. The interannual trends were given by linear regression of the average values, with the values of the slope, the standard error of estimate and the $r^2$ presented in Table 2.

Figure 5. Temporal distribution (2009-2019) of the average $C_T$, $C_{ant}$ and $C_{nat}$ in each of the layers considered for the Irminger (left plot column), Iceland (central plot column) and Rockall basins (right plot column). The average values were calculated for each cruise and layer and represented with coloured points together with their respective error bars at the time of each cruise (the method used for calculations was described in section 3.2). In the Irminger plots, the empty points represent the average values for 2019 calculated with the measured data available in the easternmost part of the basin (sampled part during this cruise), while the coloured points for 2019 represent the average values corrected with A25-OVIDE-2018 data.



The interannual trends were given by linear regression of the average values, with the values
of the slope, the standard error of estimate and the $r^2$ presented in Table 2.
Figure 6. Temporal distribution (2009-2019) of the average $pH_T$ (in situ temperature) in each
of the layers considered for the Irminger (left plot column), Iceland (central plot column) and
Rockall basins (right plot column). The average values were calculated for each cruise and
layer and represented with coloured points together with their respective error bars at the time
of each cruise (the method used for calculations was described in section 3.2). In the Irminger
plots, the empty points represent the average values for 2019 calculated with the measured
data available in the easternmost part of the basin (sampled part during this cruise), while the
coloured points for 2019 represent the average values corrected with A25-OVIDE-2018 data.
The interannual trend were given by linear regression of the average values, with the values of
the slope, the standard error of estimate and the $r^2$ presented in Table 2.
Figure 7. Temporal distribution (2009-2019) of the average $\Omega Ca$ and $\Omega Arag$ in each of the
layers considered for the Irminger (left plot column), Iceland (central plot column) and Rockall
basins (right plot column). The average values were calculated for each cruise and layer and
represented with coloured points together with their respective error bars at the time of each
cruise (the method used for calculations was described in section 3.2). In the Irminger plots,
the empty points represent the average values for 2019 calculated with the measured data
available in the easternmost part of the basin (sampled part during this cruise), while the
coloured points for 2019 represent the average values corrected with A25-OVIDE-2018 data.
The interannual trends were given by linear regression of the average values, with the values
of the slope, the standard error of estimate and the $r^2$ presented in Table 2.
**Legend for Tables**
Table 1. Metadata list of hydrographic cruises.
Table 2. Interannual trends of temperature, salinity, $C_T$, $C_{ant}$, $C_{nat}$, $pH_T$, $\Omega Ca$ and $\Omega Arag$ in each
of the layers and basins. The ratios of change were based on linear regressions applied to the
average values (as represented in Figures 4-7) and presented together with its Standard error
of estimate. The correlation coefficients $r^2$ and p-values were also provided. Values in bold
denote trends statistically significant at the 95% level of confidence.
Table 3. Temporal changes in $pH_T$ (**in $10^{-3}$ units $yr^{-1}$**) explained by fluctuations in temperature
$\left(\frac{\partial pH_T}{\partial T}\frac{\partial T}{dt}\right)$, salinity $\left(\frac{\partial pH_T}{\partial S}\frac{\partial S}{dt}\right)$, $A_T$ $\left(\frac{\partial pH_T}{\partial A_T}\frac{\partial NA_T}{dt}\right)$, and $C_T$ $\left(\frac{\partial pH_T}{\partial C_T}\frac{\partial NC_T}{dt}\right)$ in each of the layers
considered for the Irminger, Iceland and Rockall basins during the period 2009-2019. The sum
of changes explained by the individual drivers represents the calculated interannual $pH_T$
change $\left(\frac{dpH_T}{dt} \text{ calculated}\right)$, as detailed in section 2.2.5. The observed interannual $pH_T$ trends
$\left(\frac{dpH_T}{dt} \text{ observed}\right)$, shown in Figure 7 and provided in Table 2, were also added to the table for
comparison.
Table 4. Temporal changes in $\Omega Ca$ and $\Omega Arag$ (**in $10^{-3}$ units $yr^{-1}$**) explained by fluctuations in
temperature $\left(\frac{\partial \Omega}{\partial T}\frac{\partial T}{dt}\right)$, salinity $\left(\frac{\partial \Omega}{\partial S}\frac{\partial S}{dt}\right)$, $A_T$ $\left(\frac{\partial \Omega}{\partial A_T}\frac{\partial NA_T}{dt}\right)$, and $C_T$ $\left(\frac{\partial \Omega}{\partial C_T}\frac{\partial NC_T}{dt}\right)$ in each of the layers
considered for the Irminger, Iceland and Rockall basins during the period 2009-2019. The sum



of changes explained by the individual drivers represents the calculated interannual $\Omega$ change
$\left(\frac{d\Omega}{dt} \text{ calculated}\right)$, as detailed in section 2.2.6. The observed interannual $\Omega$ trends
$\left(\frac{d\Omega}{dt} \text{ observed}\right)$, shown in Figure 6 and provided in Table 2, were also added to the table for
comparison.



**Figures**
Fig. 1

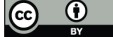

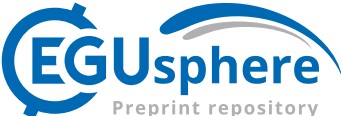

Fig. 2







Fig. 3

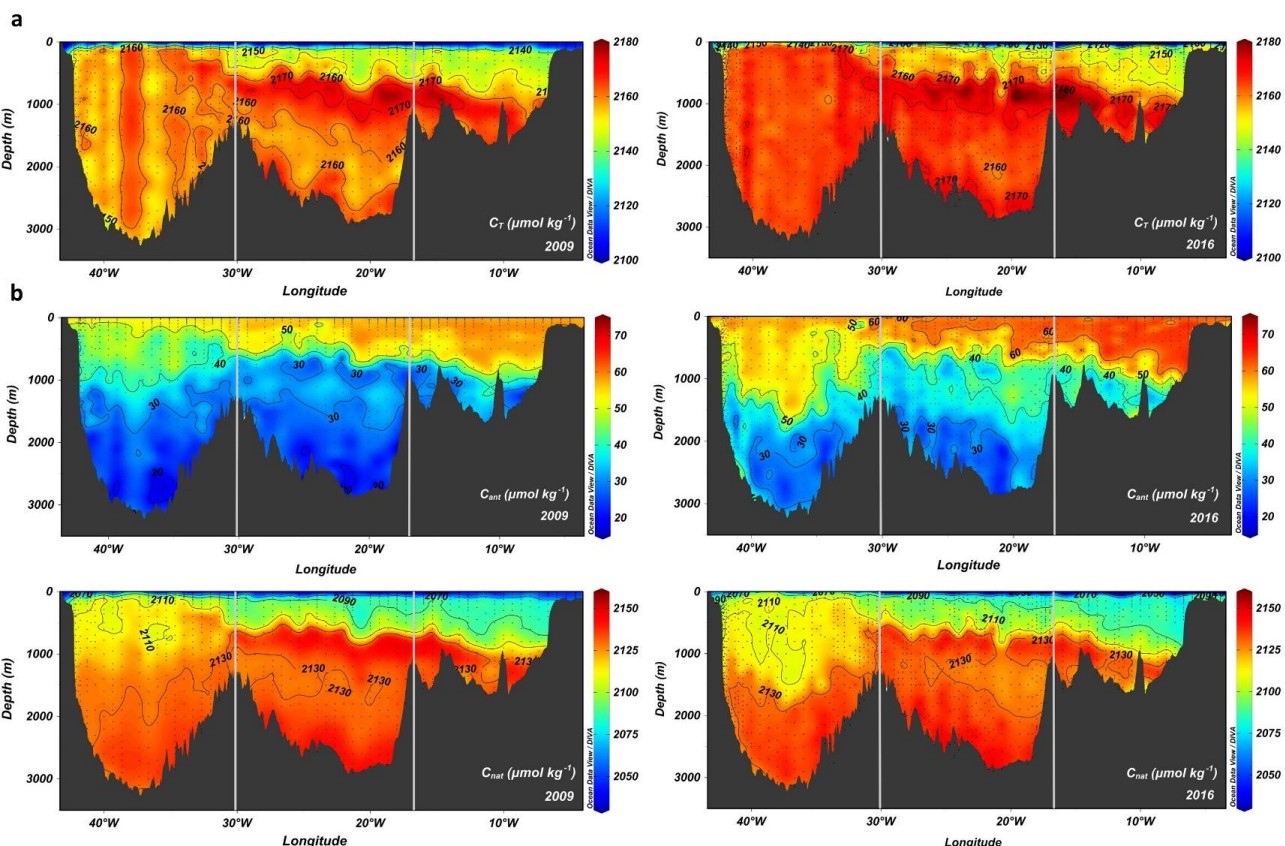



Fig. 4







Fig. 5





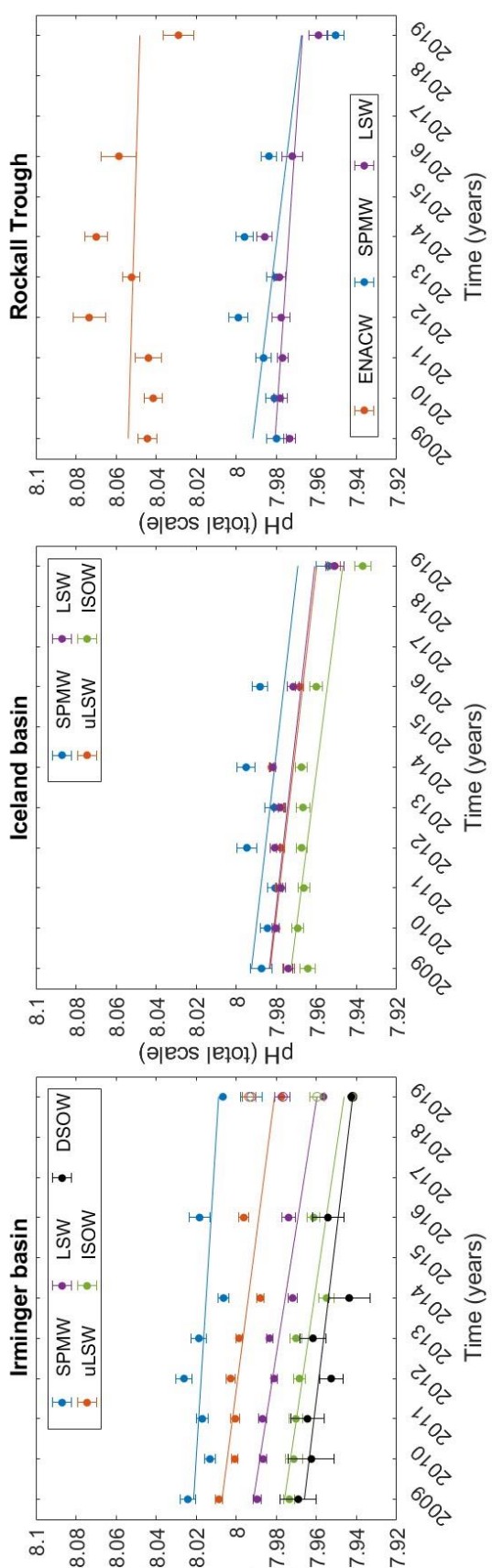

Fig. 6





Fig. 7



Table 1

| Year | Cruise ID | Date | Research Vessel (R/V) | Chief Scientist |
|------|-----------|------|-----------------------|-----------------|
| 2009 | AI28 | Aug 15-Sept 27 | Akademik Ioffe | A. Sokov |
| 2010 | AI31 | Sep 2-Sep 27 | Akademik Ioffe | A. Sokov |
| 2011 | SV33 | Sep 9-Sep 28 | Akademik Sergey Vavilov | A. Sokov |
| 2012 | AI38 | May 25-Jul 1 | Akademik Ioffe | S. Gladyshev |
| 2013 | AI41 | Jun 26-Jul 23 | Akademik Ioffe | S. Gladyshev |
| 2014 | AI44 | Jun 27-Jul 20 | Akademik Ioffe | S. Gladyshev |
| 2016 | AI51 | Jun 3-Jul 13 | Akademik Ioffe | S. Gladyshev |
| 2019 | AMK77 | Aug 8-Sep 10 | Akademik Mstislav Keldysh | S. Gladyshev |





Table 2

| Basin | Layer | Temperature | | | Salinity | | | C$_T$ | | | C$_{ant}$ | | | C$_{nat}$ | | | ph$_T$ | | | Ωca | | | ΩArag | | |
|---|---|---|---|---|---|---|---|---|---|---|---|---|---|---|---|---|---|---|---|---|---|---|---|---|---|
| | | ratio (°C yr⁻¹) | r² | p-value | ratio (psu yr⁻¹) | r² | p-value | ratio (μmol kg⁻¹ yr⁻¹) | r² | p-value | ratio (μmol kg⁻¹ yr⁻¹) | r² | p-value | ratio (μmol kg⁻¹ yr⁻¹) | r² | p-value | ratio (10⁻³ units yr⁻¹) | r² | p-value | ratio (units yr⁻¹) | r² | p-value | ratio (units yr⁻¹) | r² | p-value |
| Irminger | SPMW | -0.058 ± 0.024 | 0.60 | 0.02 | -0.006 ± 0.003 | 0.59 | 0.03 | 0.62 ± 0.23 | 0.66 | 0.02 | 0.95 ± 0.17 | 0.89 | <0.01 | -1.00 ± 0.42 | 0.60 | 0.02 | -1.25 ± 0.93 | 0.32 | 0.14 | -0.011 ± 0.006 | 0.50 | 0.05 | -0.007 ± 0.003 | 0.53 | 0.04 |
| | uLSW | -0.014 ± 0.011 | 0.30 | 0.16 | -0.002 ± 0.001 | 0.59 | 0.03 | 1.02 ± 0.18 | 0.89 | <0.01 | 1.48 ± 0.29 | 0.87 | <0.01 | -0.47 ± 0.38 | 0.28 | 0.18 | -2.62 ± 0.69 | 0.79 | <0.01 | -0.008 ± 0.005 | 0.40 | 0.09 | -0.006 ± 0.003 | 0.44 | 0.08 |
| | LSW | -0.010 ± 0.008 | 0.31 | 0.15 | -0.002 ± 0.001 | 0.50 | 0.05 | 0.98 ± 0.26 | 0.78 | <0.01 | 1.53 ± 0.23 | 0.92 | <0.01 | -0.54 ± 0.30 | 0.46 | 0.06 | -3.17 ± 0.52 | 0.91 | <0.01 | -0.014 ± 0.003 | 0.85 | <0.01 | -0.009 ± 0.002 | 0.85 | <0.01 |
| | ISOW | -0.002 ± 0.003 | 0.11 | 0.42 | 0.000 ± 0.000 | 0.00 | 0.99 | 0.90 ± 0.34 | 0.64 | 0.02 | 1.18 ± 0.29 | 0.81 | <0.01 | -0.27 ± 0.20 | 0.32 | 0.14 | -2.97 ± 0.70 | 0.83 | <0.01 | -0.010 ± 0.003 | 0.73 | <0.01 | -0.007 ± 0.002 | 0.74 | <0.01 |
| | DSOW | -0.008 ± 0.008 | 0.22 | 0.25 | 0.001 ± 0.001 | 0.43 | 0.08 | 1.32 ± 0.23 | 0.90 | <0.01 | 1.77 ± 0.32 | 0.89 | <0.01 | -0.32 ± 0.33 | 0.19 | 0.28 | -2.41 ± 0.87 | 0.67 | <0.01 | -0.004 ± 0.003 | 0.39 | 0.10 | -0.003 ± 0.002 | 0.46 | 0.07 |
| Iceland | SPMW | -0.074 ± 0.022 | 0.74 | <0.01 | -0.013 ± 0.002 | 0.89 | <0.01 | 0.85 ± 0.64 | 0.32 | 0.15 | 1.02 ± 0.31 | 0.74 | <0.01 | -0.19 ± 0.74 | 0.02 | 0.75 | -2.32 ± 1.63 | 0.34 | 0.13 | -0.016 ± 0.010 | 0.37 | 0.11 | -0.010 ± 0.007 | 0.39 | 0.10 |
| | uLSW | -0.012 ± 0.005 | 0.63 | 0.02 | -0.002 ± 0.000 | 0.76 | <0.01 | 0.68 ± 0.22 | 0.71 | <0.01 | 1.42 ± 0.38 | 0.78 | <0.01 | -0.74 ± 0.21 | 0.75 | <0.01 | -2.31 ± 1.01 | 0.58 | 0.03 | -0.009 ± 0.005 | 0.46 | 0.07 | -0.006 ± 0.003 | 0.47 | 0.06 |
| | LSW | 0.005 ± 0.003 | 0.43 | 0.08 | 0.000 ± 0.000 | 0.28 | 0.18 | 0.88 ± 0.22 | 0.80 | <0.01 | 1.18 ± 0.35 | 0.75 | <0.01 | -0.26 ± 0.26 | 0.20 | 0.27 | -2.26 ± 1.06 | 0.54 | 0.04 | -0.008 ± 0.005 | 0.41 | 0.09 | -0.005 ± 0.003 | 0.41 | 0.09 |
| | ISOW | -0.003 ± 0.006 | 0.05 | 0.61 | -0.001 ± 0.000 | 0.47 | 0.05 | 0.98 ± 0.17 | 0.89 | <0.01 | 1.20 ± 0.32 | 0.79 | <0.01 | -0.23 ± 0.21 | 0.23 | 0.23 | -2.58 ± 0.99 | 0.64 | <0.01 | -0.007 ± 0.004 | 0.42 | 0.08 | -0.005 ± 0.003 | 0.43 | 0.08 |
| Rockall | ENACW | -0.073 ± 0.061 | 0.27 | 0.19 | -0.017 ± 0.004 | 0.80 | <0.01 | 0.05 ± 0.57 | 0.00 | 0.92 | 0.85 ± 0.11 | 0.94 | <0.01 | -0.84 ± 0.50 | 0.43 | 0.08 | -0.58 ± 2.31 | 0.02 | 0.77 | -0.012 ± 0.013 | 0.18 | 0.30 | -0.008 ± 0.008 | 0.19 | 0.28 |
| | SPMW | -0.085 ± 0.019 | 0.84 | <0.01 | -0.013 ± 0.003 | 0.85 | <0.01 | 0.86 ± 0.46 | 0.48 | 0.05 | 0.87 ± 0.18 | 0.86 | <0.01 | -0.07 ± 0.59 | 0.00 | 0.88 | -2.43 ± 1.90 | 0.30 | 0.16 | -0.021 ± 0.013 | 0.38 | 0.10 | -0.013 ± 0.008 | 0.39 | 0.10 |
| | LSW | -0.020 ± 0.016 | 0.29 | 0.17 | -0.002 ± 0.001 | 0.30 | 0.16 | 0.35 ± 0.29 | 0.27 | 0.19 | 1.38 ± 0.34 | 0.81 | <0.01 | -1.05 ± 0.24 | 0.84 | <0.01 | -1.36 ± 0.97 | 0.34 | 0.13 | -0.008 ± 0.004 | 0.45 | 0.07 | -0.005 ± 0.003 | 0.45 | 0.07 |




Table 3

| Basin | Layer | $\frac{\partial pH_T}{\partial T}\frac{\partial T}{dt}$ | $\frac{\partial pH_T}{\partial S}\frac{\partial S}{dt}$ | $\frac{\partial pH_T}{\partial A_T}\frac{\partial NA_T}{dt}$ | $\frac{\partial pH_T}{\partial C_T}\frac{\partial NC_T}{dt}$ | $\frac{dpH_T}{dt}$ (obs) | $\frac{dpH_T}{dt}$ (calculated) |
|---|---|---|---|---|---|---|---|
| Irminger | SPMW | 0.91 ± 0.38 | 0.05 ± 0.02 | 0.31 ± 0.43 | -2.67 ± 0.63 | -1.25 ± 0.93 | -1.41 ± 0.85 |
| | uLSW | 0.22 ± 0.17 | 0.02 ± 0.01 | -0.10 ± 0.40 | -2.99 ± 0.53 | -2.62 ± 0.69 | -2.86 ± 0.68 |
| | LSW | 0.16 ± 0.12 | 0.01 ± 0.01 | -0.04 ± 0.39 | -2.85 ± 0.62 | -3.17 ± 0.52 | -2.72 ± 0.74 |
| | ISOW | 0.03 ± 0.05 | 0.00 ± 0.00 | -0.13 ± 0.30 | -2.38 ± 0.88 | -2.97 ± 0.70 | -2.48 ± 0.93 |
| | DSOW | 0.13 ± 0.12 | -0.01 ± 0.00 | -0.60 ± 0.18 | -3.41 ± 0.62 | -2.41 ± 0.87 | -3.90 ± 0.66 |
| Iceland | SPMW | 1.15 ± 0.35 | 0.10 ± 0.02 | 0.61 ± 0.19 | -4.14 ± 1.76 | -2.32 ± 1.63 | -2.27 ± 1.81 |
| | uLSW | 0.19 ± 0.08 | 0.01 ± 0.00 | -0.24 ± 0.45 | -2.08 ± 0.66 | -2.31 ± 1.01 | -2.12 ± 0.80 |
| | LSW | -0.08 ± 0.05 | 0.00 ± 0.00 | -0.04 ± 0.44 | -2.26 ± 0.57 | -2.26 ± 1.06 | -2.38 ± 0.72 |
| | ISOW | 0.04 ± 0.10 | 0.01 ± 0.00 | 0.12 ± 0.40 | -2.70 ± 0.43 | -2.58 ± 0.99 | -2.53 ± 0.60 |
| Rockall | ENACW | 1.13 ± 0.94 | 0.14 ± 0.04 | 0.73 ± 0.66 | -2.25 ± 1.39 | -0.58 ± 2.31 | -0.25 ± 1.80 |
| | SPMW | 1.31 ± 0.29 | 0.10 ± 0.02 | 0.47 ± 0.22 | -3.84 ± 1.23 | -2.43 ± 1.90 | -1.96 ± 1.28 |
| | LSW | 0.30 ± 0.24 | 0.01 ± 0.01 | -0.14 ± 0.37 | -0.94 ± 0.86 | -1.36 ± 0.97 | -0.76 ± 0.96 |



Table 4

| Basin | Layer | $\frac{\partial\Omega}{\partial T}\frac{\partial T}{dt}$ | $\frac{\partial\Omega}{\partial S}\frac{\partial S}{dt}$ | $\frac{\partial\Omega}{\partial A_T}\frac{\partial NA_T}{dt}$ | $\frac{\partial\Omega}{\partial C_T}\frac{\partial NC_T}{dt}$ | $\frac{d\Omega}{dt}$ (obs) | $\frac{d\Omega}{dt}$ (calculated) |
|---|---|---|---|---|---|---|---|
| Irminger | SPMW Calcite | -0.57 ± 0.24 | -0.43 ± 0.18 | 1.68 ± 2.37 | -13.35 ± 3.14 | -11.03 ± 5.57 | -12.67 ± 3.94 |
| | SPMW Aragonite | -0.49 ± 0.20 | -0.29 ± 0.12 | 1.07 ± 1.50 | -8.47 ± 1.99 | -7.17 ± 3.46 | -8.17 ± 2.50 |
| | uLSW Calcite | -0.17 ± 0.13 | -0.12 ± 0.05 | -0.46 ± 1.82 | -12.61 ± 2.24 | -8.28 ± 5.16 | -13.36 ± 2.89 |
| | uLSW Aragonite | -0.13 ± 0.10 | -0.08 ± 0.03 | -0.29 ± 1.16 | -8.03 ± 1.43 | -5.55 ± 3.21 | -8.53 ± 1.84 |
| | LSW Calcite | -0.15 ± 0.11 | -0.09 ± 0.05 | -0.17 ± 1.55 | -10.42 ± 2.27 | -13.54 ± 2.88 | -10.83 ± 2.75 |
| | LSW Aragonite | -0.11 ± 0.08 | -0.06 ± 0.03 | -0.11 ± 0.99 | -6.69 ± 1.45 | -8.65 ± 1.83 | -6.97 ± 1.76 |
| | ISOW Calcite | -0.04 ± 0.05 | 0.00 ± 0.01 | -0.44 ± 1.03 | -7.48 ± 2.75 | -10.35 ± 3.23 | -7.96 ± 2.94 |
| | ISOW Aragonite | -0.02 ± 0.04 | 0.00 ± 0.01 | -0.29 ± 0.67 | -4.84 ± 1.78 | -6.66 ± 2.04 | -5.15 ± 1.90 |
| | DSOW Calcite | -0.13 ± 0.12 | 0.03 ± 0.02 | -1.78 ± 0.52 | -9.23 ± 1.68 | -4.30 ± 2.76 | -11.11 ± 1.77 |
| | DSOW Aragonite | -0.09 ± 0.09 | 0.02 ± 0.01 | -1.16 ± 0.34 | -6.01 ± 1.10 | -3.02 ± 1.68 | -7.24 ± 1.15 |
| Iceland | SPMW Calcite | -0.88 ± 0.26 | -0.86 ± 0.16 | 3.16 ± 1.00 | -19.59 ± 8.35 | -15.77 ± 10.40 | -18.17 ± 8.42 |
| | SPMW Aragonite | -0.72 ± 0.22 | -0.58 ± 0.10 | 2.02 ± 0.64 | -12.48 ± 5.32 | -10.37 ± 6.55 | -11.77 ± 5.37 |
| | uLSW Calcite | -0.17 ± 0.07 | -0.09 ± 0.03 | -1.02 ± 1.89 | -7.98 ± 2.52 | -9.18 ± 5.11 | -9.26 ± 3.15 |
| | uLSW Aragonite | -0.12 ± 0.05 | -0.06 ± 0.02 | -0.65 ± 1.21 | -5.11 ± 1.61 | -5.92 ± 3.23 | -5.95 ± 2.02 |
| | LSW Calcite | 0.08 ± 0.05 | 0.02 ± 0.01 | -0.15 ± 1.70 | -7.92 ± 2.00 | -7.53 ± 4.64 | -7.97 ± 2.63 |
| | LSW Aragonite | 0.06 ± 0.03 | 0.01 ± 0.01 | -0.09 ± 1.09 | -5.10 ± 1.29 | -4.83 ± 2.96 | -5.12 ± 1.69 |
| | ISOW Calcite | -0.04 ± 0.10 | -0.03 ± 0.02 | 0.41 ± 1.37 | -8.38 ± 1.33 | -7.22 ± 4.34 | -8.05 ± 1.91 |
| | ISOW Aragonite | -0.03 ± 0.07 | -0.02 ± 0.01 | 0.27 ± 0.89 | -5.43 ± 0.86 | -4.72 ± 2.76 | -5.22 ± 1.24 |
| Rockall | ENACW Calcite | -0.82 ± 0.69 | -1.50 ± 0.38 | 5.16 ± 4.63 | -14.21 ± 8.78 | -11.60 ± 12.67 | -11.37 ± 9.95 |
| | ENACW Aragonite | -0.79 ± 0.66 | -1.00 ± 0.25 | 3.29 ± 2.95 | -9.06 ± 5.60 | -7.66 ± 7.96 | -7.57 ± 6.37 |
| | SPMW Calcite | -1.15 ± 0.26 | -0.82 ± 0.18 | 2.44 ± 1.15 | -18.21 ± 5.83 | -20.57 ± 13.40 | -17.74 ± 5.95 |
| | SPMW Aragonite | -0.93 ± 0.21 | -0.55 ± 0.12 | 1.56 ± 0.74 | -11.66 ± 3.73 | -13.24 ± 8.47 | -11.58 ± 3.81 |
| | LSW Calcite | -0.28 ± 0.22 | -0.10 ± 0.08 | -0.58 ± 1.57 | -3.62 ± 3.30 | -7.88 ± 4.41 | -4.59 ± 3.66 |
| | LSW Aragonite | -0.21 ± 0.16 | -0.07 ± 0.05 | -0.37 ± 1.01 | -2.33 ± 2.12 | -4.97 ± 2.82 | -2.97 ± 2.35 |



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
