# Peer review of "Ocean Acidification trends and Carbonate System dynamics"

_EGUsphere, 2024_

## Author Comment (AC1)

**Reviewer 1**

Review of the paper: egusphere-2024-1388: Ocean Acidification trends and Carbonate System dynamics in the North Atlantic Subpolar Gyre during 2009-2019, by David Curbelo-Hernández et al.

General Comment:

Quantifying and understanding the variations of anthropogenic CO2 (Cant) in the oceans is important not only to reduce the uncertainty in the present Cant inventories but also to better predict the climate/carbon coupling, i.e. how the ocean will capture atmospheric CO2 in the future. In addition the increase of Cant leads to ocean acidification and potential damage for marine species. The North Atlantic Ocean is an important CO2 sink (Takahashi et al. 2009) and this region contains high concentrations of anthropogenic CO2 (Cant) in the water column (Khatiwala et al 2013). Decadal variations of the Cant inventories were recently identified at basin scale probably linked to the change of the Atlantic Meridional Overturning Circulation (AMOC) (Gruber et al, 2019; Müller et al, 2023; Perez et al, 2024). It is worth noting that biases of AMOC in the GOBMs have been identified (Terhaar et al, 2024) that may explain differences of the Cant inventories between simulations and observations. Therefore, to better evaluate and correct current GOBMs comparisons with observed CT, AT and data-based Cant estimates are highly needed.

In this context David Curbelo-Hernández and co-authors present a detail analysis of the carbonate system (CS) changes based on 8 cruises conducted between 2009 and 2019 in the North Atlantic, here the NASPG region. After a nice introduction authors described in detail the dataset and the methods (measurements and calculations) used to investigate the temporal changes in the water column. I am convinced with all results that offer new views of the CS changes in the north Atlantic. The manuscript is clear, figures and tables adapted. The data presented here will be useful when revisiting the decadal changes of Cant inventories in the ocean (for RECCAP 3 ?), especially in the north Atlantic where decadal change of the Cant inventory could reached -1 PgC per decade (Müller et al, 2023). I support publication of this paper with minor revisions. Few comments and suggestions are listed below.

Specific comments:

C-00: Title: "Ocean Acidification trends and Carbonate System dynamics in the North Atlantic Subpolar Gyre during 2009-2019". As you investigate acidification trends in the water column (other studies analyzed this in the surface only, e.g. Lauvset et al, 2015; Leseurre et al, 2020) you may change the title: "Ocean Acidification trends and Carbonate System dynamics in the North Atlantic Subpolar Gyre water masses during 2009-2019".

The title will be updated in the new version of the manuscript.

C-01: In the Introduction, maybe recall that acidification rates were also investigated from surface data (Lauvset et al 2015; Chau et al, 2024). For example, in the Irminger Sea, Chau et al (2024) evaluated trend over 1985-2021 for pH (-0.016 ±0.001 per decade) and for War (-0.039 ±0.009 per decade).

We have added at the beginning of the second-to-last paragraph the surface OA trend for the entire North Atlantic Subpolar biome (RECCAP biome) reported by Lauvset et al., 2015) and for the Irminger and Iceland basins recently reported by Chau et al., 2024.

C-02: Line 149: Authors write: "A detailed overview and metadata of the cruises is given in Table 1."

I guess the metadata are not listed in Table 1: add a DOI or a reference for each cruise in Table 1?

The cruise reports are not published, thus cannot be referenced. However, we have completed the table with metadata information (number of sampling stations and measured variables during each cruise). A further description of the dataset is also provided in Zenodo (https://zenodo.org/records/10276222; DOI: 10.5281/zenodo.10276221).

C-03: Line 162: accuracy of ±1.5 µmol kg-1 for AT and ±1.0 µmol kg-1 for CT: this is very good. Impressive this is lower that the "climate goal" of ±2.0 µmol kg-1 (Newton et al, 2015).

That is what we get by using the full set of CRMs in each cruise.

C-04: Line 181: Are you sure that DO measurement is described in Dickson and Goyet (1994) ? Maybe refer to Dickson (1995).

Done in the updated version.

C-05: Line 382: Authors write: "The AT show a well-correlated direct relationship with salinity throughout the section (r2=0.89),…". Curiosity: authors discussed significant changes in N-AT (Figure S4), is there a detectable difference of AT/S relationship over time ?

Considering the total amount of data collected from surface-to-bottom along the entire section, there is a linear relationship between $A_T$ and Salinity with high degree of correlation ($r^2 = 0.90$ and p-value < 0.01). In the updated version of the manuscript, we provide the linear equation [$A_T$ = 54.57 (±0.36) Salinity + 396.7 (± 12.7)], that can be used to estimate the $A_T$ content along the NASPG based on salinity data with a standard error of estimate of 2.9 µmol kg$^{-1}$ (<0.1%).

Analysing fluctuations in the $A_T$/S relationship can greatly contribute to understanding the observed interannual changes in $A_T$, $NA_T$, and S. To achieve this, we examined the linear relationship between $A_T$ and S for each year of observation and plotted the slope values along with their respective standard deviations (Figure below). The linear regression indicates an increase in the $A_T$/S ratio with a statistical confidence level higher than 99%.

The interannual increase in the $A_T$/S ratio was related to the freshening (Figure 4) and the progressive increase of $A_T$-rich water inflows through upper layers (observed in the positive trends of $NA_T$ in SPMW and ENACW; Figure S4). This was likely associated with the stagnation of $A_T$-rich subtropical waters in the upper layers due to the slowdown of the NASPG since the mid-90s (e.g., Böning et al., 2006; Häkkinen and Rhines, 2004), along with changes in the spreading of waters from higher latitudes influenced by melting.

The $A_T/S$ ratio for the year 2012 was considered an outlier and excluded from the linear fitting. The high $A_T/S$ value in 2012 compared to adjacent years was explained by a combination of factors: (1) The enhanced deepening of the MLD during winter in the Irminger Sea, which promoted intense ventilation confined to the surface layer (observed in low AOU, Figure S6) and increased ocean heat loss (observed in a lower temperature signal, Figure 4), as reported by Frob et al., 2016. (2) The eastward transport likely contributed to the depletion of $A_T$ and $NA_T$ at intermediate and deeper layers in the Irminger Basin while increasing them in the Iceland and Rockall Basins. (3) Fluctuations in the spreading of ENACW into the Rockall Trough, observed in relatively low temperature and high salinity signals (Figure 4).

We have addressed the interannual variation observed for the AT/S relationship in the updated version of the manuscript (Appendix B).

[Figure]

Figure. (a) Observations of $A_T$ versus Salinity during the cruises in 2010, 2012, 2014 and 2016 plotted along with the regression lines, equations and $r^2$ (statistically significant at the 99% level of confidence). (b) Temporal variations in the $A_T/S$ relationship with error bars representing the standard deviations. The continuous red line depict the linear regression, whose slope represent the interannual change in $A_T/S$. The linear equation, $r^2$, p-value and standard error of estimate is shown in the panel. The $A_T/S$ for the cruise of 2012 was considered an outlier and excluded from the fitting. The dash red lines represent error margins of the prediction ($1\sigma$).

C-06: Line 392: Figure S2 presents the N-AT distribution for 2009 and 2016. Curiosity: Why in 2016 the N-AT distribution appeared noisy and concentrations higher in the eastern sector compared to 2009 ? Is the same was observed on other recent cruises 2014 or 2019 ?

It is true that the section for NAT during 2016 looks noisier than for 2009. It is due to the enhanced spatial resolution of the most recent cruises in comparison with the first ones. The number of sampling stations increased since 2009. The black dots, which represent data points, are closer one to each other in the panel of 2016 and thus interpolations were performed (using ODV) with higher degree of certainty than for 2009.

C-07: Line 438: Authors write: "The interannual ratios are presented along with their respective standard error of estimate and correlation factors (r2 and p-value) in Table 3 and S4.". Correct: Table 2 (not 3).

Done in the updated version.

C-08: Line 451: "SEANOE [https://www.seanoe.org/], Pascale et al., 2022". Correct name: Lherminier et al 2022.

Done in the updated version.

C-09: Line 517: Authors write: "while the Cnat experienced a slightly decrease throughout the region (Figure 5 and Table 2)." Any chance to specify the origin of this decrease ? (e.g. less export over time?)

See C-11

C-10: Line 519: Authors write: "The increase in the ventilation rates during this decade…". Which decade ? Maybe specify the period: "The increase in the ventilation rates over 2009-2019…"

Done in the updated version.

C-11: Line 521: Authors write: "explained the higher growth in Cant than expected due to the atmospheric CO2 increase." Could you recall the expected Cant trend due to the atmospheric CO2 increase? +0.5, +1, +1.5 µmol/kg/yr?

The entire second paragraph in section 4.2 was modified in the updated version of the manuscript to address the comments from C-09 to C-11. We have discussed the comment C-11 and we think that the statement "The increase in the ventilation rates over 2009-2019, shown by the negative AOU trends (Figure S6 and Table S4), explained the higher growth in $C_{ant}$ than expected due to the atmospheric $CO_2$ increase" is controversial and does not reflect what our results show. Therefore, we have decided to reformulate this part of the discussion.

What we wanted to emphasize is the relevant role of deep-water formation on modulating the $C_{ant}$ and $C_{nat}$ content. The 2009-2019 was a period of increasing ventilation in which the surface-to-bottom transport of water has enhanced. It favoured the surface waters with high $C_{ant}$ (due to the increasing uptake of atmospheric $CO_2$) and low $C_{nat}$ (due to the high biological production) to be injected into the interior ocean, where $C_{ant}$ is lower and $C_{nat}$ is higher (mainly due to the enhanced remineralization). As a result, the $C_{ant}$ has increased following quasi-linear trends throughout the layers and basins, while $C_{nat}$ has slightly decreased (mainly in the Irminger Sea due to the higher ventilation of the western NASPG). It addressed the comment C-09. The differences between years in the biological carbon pump behave as a source of variation for Cnat and explained its non-significant trends at several layers.

Here we presented the updated paragraph:

"The entrance of $C_{ant}$ through the air-sea interface and its accumulation dominated the observed increase in $C_T$ (Figure 5 and Table 2). The increase in ventilation over 2009-2019, shown by the negative AOU trends (Figure S6 and Table S4), favoured the vertical

mixing. The upper waters, due to be in contact with the atmosphere and have high biological production rates during the warm months, show high $C_{ant}$ and low $C_{nat}$ contents. The enhanced transport of upper waters toward the interior ocean explained the rapid growth in $C_{ant}$ at intermediate and deep layers. The $C_{ant}$ trends ranged between 0.85 and 1.77 µmol kg$^{-1}$ yr$^{-1}$ (statistically significant at the 99% level). They were higher than the observed on a decadal to multidecadal scale since the late 20$^{th}$ century in the Irminger and Iceland basins (0.21-0.89 µmol kg$^{-1}$ yr$^{-1}$ during 1991-2015,García-Ibáñez et al., 2016; and 0.38-1.15 µmol kg$^{-1}$ yr$^{-1}$ during 1983-2013, Pérez et al., 2021), which show the enhancement in the $C_{ant}$ accumulation on interannual scales during periods of high ventilation, as previously reported by Perez et al., (2008). The $C_{nat}$ show an inverse relationship with $C_{ant}$ at intermediate and deep layers ($r^2$>0.5; statistically significant at the 95% level of confidence) and weakly decreased across the western deep-convection NASPG (Figure 5 and Table 2). The increasing ventilation depleted $C_{nat}$ in the upper waters by transporting it toward the interior ocean. The $C_{nat}$ showed a weaker decrease at intermediate and deep layers due to the dominance of remineralization, which was not intense enough at this time of the year to neutralize the downward transport of low-$C_{nat}$ water from the surface but accounted to partially compensate for its effect. The observed variations in $C_{nat}$ between years were strongly linked with fluctuations in the biological processes explained its non-significant trends at several layers. The changes in the circulation pattern of the NASGP and thus in the horizontal advection related with the climatological forcing (Balmaseda et al., 2007; Desbruyères et al., 2013; Mercier et al., 2015; Thomas et al., 2008; Xu et al., 2013) could behave as a source of variability for both $C_{ant}$ and $C_{nat}$ and also infers differences between consecutive years".

C-12: Line 561: Authors write: "keeping approximately constant the CT (Table 3)". Correct to Table 2. Maybe recall in the text the value of the CT trend = +0.05 µmol/kg/yr in ENACW.

Done in the updated version.

C-13: Line 583: Authors write: "…where strong slowdowns in ventilation were observed from 2009 to 2010 and from 2013 to 2014, resulted in a relatively increase in Cnat and decrease in Cant observed in SPMW".  Is this signal could be associated with the low NAO index in 2010?

We updated this paragraph as follow:

"The extreme negative NAO index of 2009-2010 (Jung et al., 2011) weakened the wind forcing, which infers variability in the circulation patterns and physical properties of the surface waters, consequently reducing deep convection. This was observed in the slowdown in ventilation from 2009 to 2010 (Figure S6) in the Irminger and Iceland basins which caused a relatively increase in $C_{nat}$ and decrease in $C_{ant}$ (Figure 5)".

C-14: Line 635: Authors write: "The year-to-year variability in the biogeochemical patterns after 2012 may be attributed to the fluctuations in the spreading into the Rockall Trough of several water masses occupying different depths coming from the south and east". As there is also variability in AT and N-AT, would that be also associated to local

or regional biological processes (e.g. Cocco Blooms) and linked to the NAC transports (as discussed lines 556-559).

Our results show that the ENACW in the Rockall Trough was well-oxygenated (low AOU values; Figure S4) but the ventilation was highly variable over time (several differences were found for AOU between consecutive years). This likely introduced differences in the biological carbon pump and in the horizontal and vertical advection which collectively introduced heterogeneities in the temporal distribution of $A_T$, $NA_T$, $C_T$ and $NC_T$ (Figure 5, S4 and S5). We have added the following statement to the updated version of the manuscript: "The heterogeneities in the ventilation of the ENACW between consecutive years likely influenced the biological and advective patterns, introducing differences in $A_T$, $NA_T$, $C_T$ and $NC_T$ (Figure 5, S4 and S5)".

C-15: Line 672: Authors write: "The substantial variability introduced by these processes made it difficult to discern the pattern of acidification and its drivers on an interannual scale in the shallow Rockall Trough. Therefore, long-term monitoring and the development of multidecadal-scale studies are required in this area to derive significant conclusions." I agree that maintaining long-term monitoring is important and it is often difficult to detect and explain the pH variations: however it seems that, and opposed to other regions, the Cant trend in the ENACW is relatively well estimated (quasi-linear) in the Rockall Trough (Figure 5) with value of 0.85 ±0.11 µmol/kg/yr and that Cnat decrease is also well evaluated (-0.84 ± 0.50 µmol/kg/yr). For this water mass would it be possible to suggest some process that explain the Cnat decreasing trend since 2009 (physics and/or biology ?).

We have modified this paragraph in the updated version of the manuscript. We decided to remove that statement as is not appropriate in this part of the Manuscript. The long-term monitoring is important in regions which present high variability at different timescales such as the Rockall Trough. It is required to evaluate with higher certainty (with and statistical level of confidence higher than the 95%) the variations in the physical and biogeochemical seawater properties. As the differences between consecutive years encountered for MCS properties over 2009-2019 have behaves as a source of uncertainty at some layers as for example the ENACW, we would like to highlight the need of continue monitor and develop observation-based studies in such crucial areas, as we detailed in the conclusions.

In the updated paragraph (section 4.3), we suggest that the observed variability in the MCS through the ENACW is linked with changes in the advection, which modified the ventilation and spreading of water masses with different physical and biogeochemical patterns, and in the biological processes. Holliday et al., (2020) reported the freshening of the eastern NASPG led the weakening in the poleward advection of saline waters and strengthening in the eastward recirculation of freshwater during 2012-2016. The observed decrease in salinity (Figure 4) and its impact on $A_T$ compared to $NA_T$ (Figure S4) reflect this behaviour. The decrease in $C_{nat}$ offsetting the increase in $C_{ant}$ was also discussed: the depletion in $C_{nat}$ could be related to the increasing biological uptake (Ostle et al., 2022) and physical and biogeochemical changes driven by fluctuations in the lateral advection (Holliday et al., 2020).

The updated paragraph is presented here:

"The upper waters of the Rockall Trough presented the maximum $pH_T$ throughout the transect (8.02-8.08 units). The observed strong $pH_T$ fluctuations between years related with interannual changes in the NAC do not allow to discern trends with a statistically interval of confidence equal or higher than the 90%. The interannual decrease in $pH_T$ in the ENACW (~0.001 units $yr^{-1}$) was half than the observed along southernmost transects in the Rockall Trough between 1991 and 2010 (~0.002 units $yr^{-1}$, McGrath et al., 2012a). The temporal distribution of the average $pH_T$ (Figure 6) highly influenced by the high-ventilation (seen in minimum AOU values highly variables between years and which tend to decrease with 99% statistical confidence; Figure S6 and Table S4) allow to discern two periods: the approximately constant ventilation rates keep a steady state in terms of $pH_T$ during 2009-2011, while the progressively renewal and oxygenation of subsurface waters after 2012 (and peaking in this year) increase the $pH_T$. The renewal of waters in the shallow Rockall Trough, in contrast with the westernmost NASPG, was not primarily driven by vertical but by lateral advection. The modifications of the ENACW through air-sea exchange and mixing with adjacent waters modulated its properties at different time scales (Holliday et al., 2000) and caused the observed variations in the MCS. The variations in $pH_T$ between consecutive years after 2012 may be attributed to the fluctuations in the spreading into the Rockall Trough of several water masses occupying different depths coming from the south and east (Ellett et al., 1986; Pollard et al., 1996). Holliday et al., 2020 reported the reduction in the spreading of saline subsurface waters from subtropical latitudes and diversion of Arctic freshwater from the western boundary into the eastern NASPG during 2012-2016. The subsequent freshening of the ENACW compensated for the increase in $A_T$ expected without the effect of salinity (see in the decreasing $A_T$ against the increasing $NA_T$; Figure S4 and Table S4) and weakened the increase in $C_T$ expected due to poleward advection (see in the slowdown in the rise of $C_T$ in comparison with those of NCT; Figure 5 and S5 and Table 2 and S4). The $C_T$ remains approximately constant (Figure 5 and Table 2) due to the increase in $C_{ant}$ (0.85 ± 0.11 µmol $kg^{-1}$ $yr^{-1}$; p-value < 0.01) was neutralized by the decrease in $C_{nat}$ (-0.84 ± 0.50 µmol $kg^{-1}$ $yr^{-1}$; p-value < 0.1). These findings suggest that the atmospheric $CO_2$ invasion was offset by the growing phytoplankton biomass favouring its biological uptake (Ostle et al., 2022) and the weakening transport of remineralized and saline water from the south (Holliday et al., 2020), thus compensating the acidification of the ENACW".

C-16: Line 697: Authors write: "the positive NAT trends encountered in the upper layers lead a rise in pHT, while the diminished NAT contributed to decrease the pHT toward the interior ocean." Is the N-AT trend could be related to change in Cocco blooms distributions or the signal is too low to interpret the link with the various biological processes in the north atlantic (Ostle et al, 2022) ?

The many processes involved in the distribution of $NA_T$ difficult to assess its trend. The evaluation of the $NA_T$ drivers required to consider the biogeochemical processes influencing the bicarbonate-carbonate equilibrium in seawater. The variations in $NA_T$ could be related to changes in biological production, but its direct influence is not significant. However, it has an indirect effect through the production/decomposition of organic matter as well as through its influence on biogeochemical cycles (i.e. nutrients). The variations in pH have an important role by changing the fundamental chemical equilibrium in seawater: the acidification favour the dissolution of carbonates, which increases $NA_T$. Lastly, the freshwater inflows with high levels of bicarbonate could increase $NA_T$.

The positive $NA_T$ trends encountered in subsurface layers (SPMW and ENACW) could be related to the diminishing in $pH_T$, which favour the dissolution of carbonates, combined with increasing biological production reported for upper layers across the NASPG (Ostle et al, 2022). It contrasts with the constant to weakly decreasing $NA_T$ trends at intermediate and deep layers, in which the accelerated acidification was compensated by the dominance of remineralization processes over lower biological uptake. The $NA_T$ trends were not-statistically significant in most of the layers and basins due to the year-to-year differences in the processes involved in its variability. This statement was including in this part of the discussion.

The increasing phytoplankton biomass and subsequent enhancing in primary production reported for the NASPG by Ostle et al, (2022) was evidenced in the decrease in $C_{nat}$, although the effect of advection also introduced variations. It has been discussed at some points of the new version of the manuscript.

C-17: Line 742: Figure 7, Table 2: The trend of War in the Irminger Sea of around -0.07 per decade seems high compared to that deduced from reconstructed products (e.g. -0.039 ±0.009 per decade, Chau et al, 2024). Could that be discussed ? Would that be explained by seasonal difference of the trends (e.g. Leseurre et al, 2020) ?

Chau et al, 2024 highlighted the uncertainty in their own pH and $\Omega_{Arag}$ estimations in the Irminger and Iceland basins (also for the estimations performed by Bates et al., 2014) due to low data-sampling frequency at their monitoring sites. We have updated this part of the discussion by including these statements: "…The $\Omega_{Arag}$ trend estimated for SPMW in the Irminger basin (-0.007 ± 0.003 units yr$^{-1}$) is consistent with that reported for surface waters by Bates et al., (2014) over 1983-2014 (-0.008 ± 0.004 units yr$^{-1}$) and fall within the range of those estimated during summer by Leseurre et al., 2020 over 2008-2017 (-0.005 ± 0.001 units yr$^{-1}$). Chau et al., 2014 recently deduced from reconstructed products a slower decrease (-0.004 ± 0.001 units yr$^{-1}$), highlighting the large uncertainty in the estimations of interannual trends for pH and $\Omega_{Arag}$ across the NASPG due to the low-data sampling frequency at their monitoring sites…"

C-18: Line 811: Authors write: "The driver analysis exhibited the strongest interannual decrease in $\Omega$ in the upper layers governed by the uptake of Cant weakly compensated by the increase in NAT and favoured by the cooling and freshening." Here, again, you identify that the N-AT increase explains part of the W changes: what is the process associated to this variations (Cocco bloom ?) ?

We explained it in the updated version of the manuscript and provided a detailed description of the processes involved in the variation of $NA_T$ to response comment C-16. We have added the following statement in section 4.4: "…The interannual increase in $NA_T$ in upper layers could be related to the diminishing in $pH_T$, which favour the dissolution of carbonates, combined with increasing biological production reported for upper layers across the NASPG (Ostle et al, 2022). It contrasts with the constant to weakly decrease in $NA_T$ at intermediate and deep layers, in which the accelerated acidification was compensated by the dominance of remineralization processes over lower biological uptake. Consequently, the positive $NA_T$ trends encountered in the upper layers lead a rise in $pH_T$, while the diminished $NA_T$ contributed to decrease the $pH_T$ toward the interior ocean …".

C-19: Line 816: Authors write: "The progressive reduction in ΏArag is driving a long-term decrease in the depth of the aragonite saturation horizon (ΏArag=1) by 80-400 m since the preindustrial era". Curiosity: could you show a section (or profiles) of preindustrial War calculated with Cnat to compare with modern values (to add in Supp Mat ?).

We agree that this can greatly enrich this section of the discussion. We have computed the preindustrial ΏArag and ΏCa with the $CO_{2sys}$ programme run in the MATLAB software and based on $C_{nat}$. The vertical sections of preindustrial ΏArag and ΏCa are depicted along with those for the cruises of 2009 and 2016 in the updated version of the Supplementary Material (Figure S3). Additionally, the isolines for the aragonite saturation horizon for preindustrial and present times are remarked in the new vertical sections.

The updated figure and capture are shown here:

[Figure]

"Figure S3. Water-column distribution along the longitudinal transect of (a) ΏArag and (b) ΏCa for preindustrial times and for the cruises of 2009 and 2016. The preindustrial ΏArag and ΏCa values were computed with the $CO_{2sys}$ programme (Lewis and Wallace, 1998) run with the MATLAB software (van Heuven et al., 2011; Orr et al., 2018; Sharp et al., 2023) using as input $CO_2$ system variables the $A_T$ and the $C_{nat}$. In panels a.2 and a.3, the highlighted continuous black isolines represent the aragonite saturation state horizon during the cruises of 2009 and 2016, respectively, while dash isolines show the aragonite saturation state horizon in preindustrial times. The vertical white lines show the limits between basins".

We also added the following sentence to the discussion (section 4.5):

"The vertical section of $\Omega_{Arag}$ in Figure S3 shows the shallower aragonite saturation horizon during 2009 and 2016 compared to preindustrial times".

C-20: Line 834: Authors write "In fact, several studies reported that CWC ecosystems are anticipated to be among the first deep-sea ecosystems to experience acidification threats". Maybe refer to Gehlen et al (2014) (and also in the introduction).

Done in the updated version.

C-21: Line 856: Authors write "The observational period is relatively short to quantify long-term trends and to formulate significant future projections." Maybe delete here, as this was also written on line 840: "Despite the observational period is relatively short to quantify long-term trends and to formulate significant future projections,…".

Done in the updated version.

C-22: 900: "to favour the entrance of Cant in intermediate and deep-layers and this its acidification,"

It was a typo and was removed

C-23: Conclusion: Maybe reduce the numbers of values listed in the conclusion (already listed in the MS).

Done in the updated version.

Figures:

C-24: Figures: check the color code for blind (see accepted colors for the journal Biogeoscience)…

We checked the readability of our figures and maps by readers with color vision deficiencies using the Color Blindness Simulator (Coblis) provided by the journal Biogeosciences. We have observed that the color scheme used for the vertical sections in Figures 1, 2, 3, S2, and S3 effectively highlights the contrasts between maximum and minimum values, allowing for a clear identification of the vertical gradient observed for each variable. Additionally, we included contour lines or isolines to facilitate the interpretation of the results and enhance the understanding of the distribution of intermediate values, where the color scheme introduces some brightness.

In contrast, the color scheme used for Figures 4, 5, 6, 7, S4, S5, and S6 may be confusing for those with dichromatic vision (protanopia and deuteranopia). We have adjusted the hues of the cool colors (violet and green) to enhance the contrast with the warm colors (orange and red) for individuals with protanopia and deuteranopia. Additionally, we have replaced the orange color with red in the uLSW layer to increase contrast with the other colors used in the Irminger and Iceland basins. We hope these changes will improve the accessibility of the figures and enhance the interpretation and impact of the results.

C-25: For Figures 4, 5, 6, 7, a suggestion: change the color code for ENACW (i.e. different than for uLSW)

Done in the updated version.

C-26: In figure 5 the CT and Cant concentrations in ENACW were low in 2014. I think this is not discussed. Any idea to explain this anomaly ? A strong bloom in 2014 in the eastern NASPG or a shift in the data for this cruise ?

During 2014, the ENACW presented low $C_{nat}$ which relatively decrease the $C_T$ (Figure 5) and high $A_T$ and $NA_T$ (Figure S4). These changes occurred in phase with the warming observed from 2012 to 2014 (Figure 4) and likely indicated that the increase in carbonate and bicarbonate concentrations rising $A_T$ and $NA_T$ was compensated by the depletion in dissolved $CO_2$. The relatively high temperature and $NA_T$ in 2014 indicates an improved spreading of subsurface waters from subtropical latitudes into the Rockall Trough. The enhanced biological production in these waters, together with the reduction in solubility due to warming which favour the $CO_2$ evasion to the atmosphere, account for decreasing $C_{nat}$ and thus $C_T$.

This explanation was added to the discussion in the new version of the manuscript.

References:

C-27: Friedlingstein et al, 2022: add full reference, journal, doi

C-28: Pascale, L., 2022: change to

Lherminier P., Perez, F. F., Branellec, P., Mercier, H., Velo, A., Messias, M. J., Castrillejo, M., Reverdin, G., Fontela, M., Baurand, F. (2022). GO-SHIP A25 - OVIDE 2018 Cruise data. SEANOE. https://doi.org/10.17882/87394

References were corrected in the updated version of the manuscript.

;;;;;;;;;;;;;; References added in this review not cited in the MS

Chau, T.-T.-T., Gehlen, M., Metzl, N., and Chevallier, F.: CMEMS-LSCE: a global, 0.25°, monthly reconstruction of the surface ocean carbonate system, Earth Syst. Sci. Data, 16, 121–160, https://doi.org/10.5194/essd-16-121-2024, 2024.

Dickson, A. D. 1995. Determination of dissolved oxygen in sea water by Winkler titration. WOCE Operations Manual, Part 3.1.3 Operations & Methods, WHP Office Report WHPO 91-1.

Gehlen, M., Séférian, R., Jones, D. O. B., Roy, T., Roth, R., Barry, J., Bopp, L., Doney, S. C., Dunne, J. P., Heinze, C., Joos, F., Orr, J. C., Resplandy, L., Segschneider, J., and Tjiputra, J.: Projected pH reductions by 2100 might put deep North Atlantic biodiversity at risk, Biogeosciences, 11, 6955-6967, 10.5194/bg-11-6955-2014, 2014.

Lauvset, S. K., Gruber, N., Landschützer, P., Olsen, A., and Tjiputra, J.: Trends and drivers in global surface ocean pH over the past 3 decades. Biogeosciences, 12, 1285-1298, doi:10.5194/bg-12-1285-2015, 2015

Lherminier P., Perez, F. F., Branellec, P., Mercier, H., Velo, A., Messias, M. J., Castrillejo, M.,Reverdin, G., Fontela, M., Baurand, F. (2022). GO-SHIP A25 - OVIDE 2018 Cruise data.SEANOE. https://doi.org/10.17882/87394

Müller, J. D., Gruber, N., Carter, B., Feely, R., Ishii, M., Lange, N., et al.: Decadal trends in the oceanic storage of anthropogenic carbon from 1994 to 2014. AGU Advances, 4, e2023AV000875. https://doi.org/10.1029/2023AV000875, 2023

Newton, J.A., Feely, R. A., Jewett, E. B., Williamson, P. and Mathis, J.: Global Ocean Acidification Observing Network: Requirements and Governance Plan. Second Edition, GOA-ON, https://www.iaea.org/sites/default/files/18/06/goa-on-second-edition-2015.pdf, 2015.

Ostle C., P. Landschützer, M. Edwards, M. Johnson, S. Schmidtko, U. Schuster, A. J. Watson and C. Robinson, 2022. Multidecadal changes in biology influence the variability of the North Atlantic carbon sink. Environ. Res. Lett. 17, 114056, DOI : 10.1088/1748-9326/ac9ecf

Terhaar, J., Goris, N., Müller, J. D., DeVries, T., Gruber, N., Hauck, J., et al. (2024). Assessment of global ocean biogeochemistry models for ocean carbon sink estimates in RECCAP2 and recommendations for future studies. Journal of Advances in Modeling Earth Systems, 16, e2023MS003840. https://doi.org/10.1029/2023MS003840

---

## Author Comment (AC2)

**Review 2**

Review of the manuscript egusphere-2024-1388 «Ocean Acidification trends and Carbonate System dynamics in the North Atlantic Subpolar Gyre during 2009-2019» by Curbelo-Hernández et al. submitted for discussion in Biogeosciences

**General comment :**

This manuscript reports a detailed analysis of carbonate chemistry parameters measured along the entire water column of a transect in the North Atlantic South Polar Gyre during 8 oceanographic cruises conducted between 2009 and 2019. The manuscript gives a nice description of this remarkable dataset. The dataset is used to study the trends of ocean acidification in this oceanic area with a particular focus on the spatial variability between three defined oceanic provinces (The Irminger Basin, the Iceland Basin and the Rockall Trough) and the different water layers encountered in the water column. Due to its importance as an oceanic carbon sink, the north Atlantic basin has been very intensively studied in the last decades. However this study is a significant contribution for (at least) three reasons : (1) it presents a new dataset, (2) it focuses on the last decade (2009-2019) and it (3) shows some original results in the eastern part of the basin. The authors have made a detailed literature review in order to put this study in context. The measurements and calculation methods are carefully described. The figures and tables are of good quality. The manuscript is well written and the results are convincing. However the manuscript is very long. This is certainly due to the fact that it is at the same time a "data paper" and a "scientific paper". My point is not to say that it should be split in two (I appreciate having all the information in one manuscript) but I believe that it could be more synthetic in some parts. I would be glad to support the publication of this manuscript after some revisions (which I believe to be minor). My major concerns (detailed in the following sections) are related to the structure of the manuscript and some methodological points.

Thank you very much for your thoughtful and constructive feedback on our manuscript. We greatly appreciate your recognition of the significance of our study, particularly in presenting a new dataset, focusing on the most recent decade, and revealing original results in the eastern North Atlantic basin. We acknowledge your point regarding the manuscript's length and the balance between being both a "data paper" and a "scientific paper." We have carefully considered your suggestions, which have contributed to enhancing the quality and reliability of the manuscript. Below, we provide a point-by-point response to each of your comments.

**Specific comments :**

Section 2.2.5 and 2.2.6: It's the same method that is used to deconvolute the drivers of the pH and omega trends. This could go into one section called "deconvolution of the trends". The beginning of section 2.2.6 on the calculation of the omega values could be added to section 2.2.2 "computational methods".

This restructuring has substantially improved the organization and readability of the methodology in the updated version of the manuscript.

On section 2.2 « Data Processing » : I am surprised about the fact that a huge amount of work has been done to compare the measured variable to variable estimated with CANYON-B but I haven't found a simple "internal consistency test" of the three measured variables of the carbonate system. Following the table S2, a table could give some basic statistics about the difference "pHT measured – pHT calculated from AT/CT

", "AT measured – AT calculated from pHT/CT" or "CT measured – CT calculated from AT/pHT" in the cases where all three variable were measured.

We conducted an extensive and detailed evaluation of the internal consistency of our observations. The measured variables were compared for each cruise with canyon-estimated and CO2sys-computed data, as you suggested. Initially, we included only comparisons with CANYON-B estimates in the table, but following your recommendation, we recognized the importance of also including comparisons with computed variables to further reinforce the reliability of our observations and results. We have now included these basic statistics on the differences between measured and computed pH, AT, and CT in Table S2. Additionally, we have added the following sentence in section 2.2.2: "An internal consistency test was conducted on the three measured MCS variables. The measured variables were compared with canyon-estimated and $CO_{2SYS}$-computed variables. The average differences and standard deviations were summarized in Table S2 and ensure the consistency of the observations".

On section 2.2.4 « Water mass characterization » : This section presents mostly results about the different water masses encountered during the cruises. Most of this section should be added (maybe in a more synthetic way ) into section 3.1 « Physicochemical characterization of the water column ».

The titles used for these two subsections may be confused and not fully reflect their content. The title of Section 2.2.4 was updated to "Hydrographic Characterization". This subsection intend to identify the main basins and water masses across the NASPG. This subsection does not present results directly aligned with the research objectives of the article, but describes the principal water masses occupying the Irminger, Iceland, and Rockall basins. It also explains how were delineated the layers by potential density isopycnals following previous studies in the area. This delineation is crucial for the subsequent calculation of means and trends in each basin and water mass. Therefore, we believe this subsection would be more appropriately placed within the methodology section.

The title of subsection 3.1 was removed as detailed in the response to the following comment. The content of the updated section 3 differs from that of subsection 2.2.4, making their unification challenging. Section 3 presents the spatio-temporal distribution of physicochemical variables once the hydrographic characterization was performed, which will be discussed in detail in Section 4.

On section 3.2. " Temporal evolution of the physicochemical properties". There are no real results in this section, most of the information here is more relevant to the method "section".

It is true. Our initial idea was to divide the Results section into two subsections: one focused on explaining the distribution of variables in the water column and another addressing temporal changes. The second subsection was intended to present the trends that would be discussed in depth in Section 4, but it included extensive information related to trend calculation and statistical analysis. We have considered your suggestions and moved this information to a new subsection in the Methodology titled "Data Adjustment for Trends Computation." Additionally, we have removed the titles of the two subsections in the Results section and unified them in the revised version of the manuscript.

On section 4.2 and Appendix B: I don't really understand the reasons to give the information of the CT and AT trends in appendix. I would suggest putting this information directly in the main text.

Initially, this information from Appendix B was incorporated into Section 4.2. To synthetize this section and reduce the overall text length and considering its lesser relevance to the main results discussed in this subsection and the central ideas being conveyed, these paragraphs were moved to Appendix B and considered supplementary to the discussion performed in section 4.2. Based on your feedback, we have decided to reintegrate the paragraph focused on CT and NCT trends into section 4.2, while trends in AT and NAT, which are complementary results that support the discussion but fall outside the article's primary objectives, will remain in the appendix.

Sections 4.3 "Acidification trends", section 4.4 "Drivers pH" and section 4.5 " Interannual changes in …" are very long. I really believe that these three sections could be synthesized into one single section. This is not an easy task but the drivers of the pH trends and the omega trends are mostly the same (I am of course aware that some differences exist) and could be treated in a more synthetic manner.

We have carefully considered this comment and discussed various possible changes to these three sections of the discussion. Our goal was to ensure that the key results and ideas are clearly conveyed while maintaining readability for the reader. We agree that this was not an easy task, as there was a risk of omitting relevant results and presenting the information in a disorganized manner. Additionally, these sections had already been revised according to suggestions from Reviewer 1.

Due to the various topics addressed in this part of the discussion, we believe that the best approach is to separate the text into subsections. Therefore, we have considered both the titles and content of these sections. In the revised manuscript, we have included the following subsections:

4.3. Acidification trends: this section has been condensed, incorporating some modifications based on the suggestions from Reviewer 1. We addressed here the $pH_T$ trends encountered in the different layers and basins, which were compared to previous studies.

4.4. Interannual changes in $\Omega_{Ca}$ and $\Omega_{Arag}$: We considered important to separate this section from the previous one, as it specifically addresses the effects of OA on individual organisms, ecosystems, and biological processes through the analysis of changes in $\Omega$. In this subsection, we removed the analysis of the drivers of $\Omega_{Ca}$ and $\Omega_{Arag}$ that was included in the corresponding part of the text in the previous version of the manuscript.

4.5. Processes controlling OA and $\Omega$ trends: After analysing the interannual trends in $pH_T$ (subsection 4.3) and $\Omega_{Ca}$ and $\Omega_{Arag}$ (subsection 4.4), we provided in this subsection a detailed analysis of the processes driving their trends quantifying its contributions. The combination of the analysis of drivers for $pH_T$ and $\Omega$, performed by following your suggestion, significantly aids in synthesizing this part of the discussion, making it more concise and easier to understand.

Conclusion: This is a really long conclusion! I would suggest just giving the main messages in conclusion and It could certainly stand in 20 to 30 lines.

We agree that the conclusion was somewhat lengthy, as also noted by Reviewer 1. To enhance readability and ensure that the key messages of this research are effectively

conveyed and better assimilated by readers, we have condensed the conclusion. We have done so by avoiding the inclusion of information already covered in the discussion section, in line with your recommendations and those provided by Reviewer 1.

The Conclusion was updated as follow:

"This research has evaluated the interannual changes in the basin-wide MCS dynamics along the NASPG during 2009-2019. Despite the observational period is relatively short to quantify long-term trends and to formulate significant future projections, the finding has allowed to evaluate the ocean response, in terms of MCS dynamics and on an interannual scale, to changes in deep-water convection and to isolate events affecting the physical patterns. The assessment of OA within the Irminger and Iceland basins was enhanced by supplying novel data and trends spanning a decade in which the physical patterns reversed. Additionally, the study provides an unprecedent analysis of the physico-chemical variations in the Rockall Trough, which is crucial for the assessment of the entire longitudinal span of the NASPG. It facilitates a more accurate understanding of the mechanisms dictating basin-scale acidification processes and advances our understanding of OA in the North Atlantic and Global Ocean.

Overall, the entrance and accumulation of Cant and interannual acidification trends were strongly affected by the cooling, freshening and enhancement in the oxygenation during this decade. The longitudinal span of the NASPG and the differences in circulation patterns, water masses and bathymetry behaved as a source of spatio-temporal variability. The interannual acidification trends of the main water masses across the NASPG ranged between 0.0006-0.0032 units yr-1 and caused a decline in the ΩCa and ΩArag of 0.004-0.021 and 0.003-0.013 units yr-1, respectively. The convective processes increased the accumulation rates of Cant in the interior ocean by 50-86% and accelerated the acidification rates by around 10% compared to previous decades in the Irminger and Iceland basins. The shallower hydrography of the Rockall Trough and the poleward circulation patterns accounted for differences in the acidification rates respect to surrounding waters.

The Cant-driven increase in NCT was found to govern the acidification of the NASPG with contributions exceeding 60%. The combined effect of the decreasing temperature, salinity and NAT neutralized close to one-half of the acidification along the entire longitudinal span of the SPMW. The enhanced deep-water ventilation in the western NASPG slowdown the cooling and freshening toward the interior ocean, weakening the physical counterbalance of acidification.

The present investigation emphasizes the progressively increase in the uptake and accumulation of Cant and subsequent acceleration of OA along the NASPG. Novel data and results provided could be compared with other repeated hydrographic section data at mid and high latitudes in the North Atlantic, such as the A02, A25, AR07E and AR28 framed in the GO-SHIP program, as well as used in conjunction to develop future investigations. Additionally, they contribute to the improvement of the projections pertaining to the future state of the oceans run by models and forecast. Considering the important variability in the mechanism controlling the distribution of the physico-biogeochemical properties and particularly the OA in the North Atlantic, this research aims to highlight the necessity of continue monitoring and sampling the whole water column through repeated hydrographic sections, especially through the highly variable but less assess easternmost part."

**Technical corrections**

L54 : What is meant by « conservative scenario » ?

In this context, we refer to the IPCC's Representative Concentration Pathways (RCPs) scenarios (Van Vuuren et al., 2011; Moss et al., 2010), which project various future trajectories of greenhouse gas concentrations. By "conservative scenario," we intended to refer to the RCP2.6 scenario, which is based on lower future $CO_2$ emissions and predicts more moderate impacts compared to more aggressive emission pathways.

We have revised this part of the introduction and we now discuss the RCP2.6 scenario as the more optimistic projection and the RCP8.5 scenario as the more pessimistic one. We also cited IPCC AR5 and AR6 reports: "According to the IPCC's Representative Concentration Pathways (RCPs) scenarios (Van Vuuren et al., 2011; Moss et al., 2010), which project various future trajectories of greenhouse gas concentrations, the model projections estimate a potential pH decrease of 0.3–0.4 units by the end of the century under the RCP8.5 scenario, which assumes continued high $CO_2$ emissions. In contrast, the most conservative RCP2.6 scenario, which includes significant emission reductions, anticipates a pH drop of 0.2–0.3 units (IPCC 2013 and 2021)".

L55 : Please define what is meant by « Anthropogenic carbon » before using it for the first time.

We added the definition given by Sarmiento et al., 1992: "fraction of inorganic carbon resulted from human emissions".

Sarmiento, J. L., Orr, J. C., and Siegenthaler, U. (1992). A perturbation simulation of CO2 uptake in an ocean general circulation model. Journal of Geophysical Research: Oceans, 97(C3), 3621-3645.

L84 : Missing reference "Gonzalez-Davilla and Santana-Casiano 2023"

It was included in the reference list.

L133 : The concept of « hydrographic CLIVAR 59.5°N Section » is not clear. Is this an historical WOCE section ?

The hydrographic section is part of the World Climate Research Programme (WCRP) within the framework of the CLIVAR (Climate and Ocean: Variability, Predictability and Change) project. We updated the manuscript with the following statements: "Data were collected from eight summer cruises conducted along the transverse hydrographic section at 59.5ºN between 2009 and 2019 (Daniault et al., 2016; Gladyshev et al., 2016b, 2017, 2018; Sarafanov et al., 2018). This repeat section is part of the World Climate Research Programme (WCRP) within the framework of the CLIVAR (Climate and Ocean: Variability, Predictability and Change) project and covers the length of the Subpolar North Atlantic between Scotland and Greenland (4.5-43.0ºW), crossing the Irminger and Iceland basins and the Rockall Trough (Figure 1)"

L160 : What is meant by « in-situ calibrated » ?

We have removed "in situ" from this sentence for a better understanding. We wanted to say that the VINDTA was systematically on-board calibrated using CRMs.

L170-173 : This point is confusing. An uncertainty of 0.0047 pH Units does not necessarily correspond to a systematic bias. Please clarify this point?

We have reformulated this point. The uncertainty of 0.0047 pH units reported by DelValls and Dickson (1998) was identified as an average difference rather than a systematic bias.

This value was derived from comparing pH values calculated using the pK* values of m-cresol purple with those obtained from more accurate pH determinations performed by DelValls and Dickson (1998).

Following their recommendations, we applied the correction of +0.0047 units to our pH measurements to address this identified discrepancy and align the pH values with those determined from the more accurate pK* values. While the uncertainty itself does not directly indicate systematic bias, applying this correction helps ensure that our pH data are consistent with the updated pK* values and thus improves the accuracy of our measurements. We have included this adjustment to reflect the more reliable pK* values and maintain consistency with the literature recommendations. In addition, after applying this correction, we got pH values with a negligible difference (0.0002 units) compared to those estimated by CANYON-B (this point was added to subsection 2.2.1), which infers confidence to the correction applied.

L199 : CANYON-B is trained and validated using GLODAPv2 data. Argo profiles are just used for comparison but not for formal validation.

The statement was updated in the new version of the manuscript: "This neural network is trained on and validated against bottle data from GLODAPv2 and recent GO-SHIP profiles and compared with sensor data from Argo floats".

L372-373 : This sentence is confusing. This section is not devoted to evaluating the temporal changes between 2009 and 2016.

In the updated version of the manuscript, we revised the beginning of this paragraph to clarify our intent. Our goal was to highlight that Figures 2, 3, S2, and S3 allow for a comparison of the surface-to-bottom distribution of variables between the 2009 cruise (left panels) and the 2016 cruise (right panels). After revision, we simplified this idea to: "The vertical distribution of the physical and biogeochemical variables is depicted for the cruises of 2009 and 2016 in Figures 2, 3, S2 and S3".

L550 – 553 : This sentence is not completely clear to me. Why is the limitation of the ventilation to the subsurface water related to an enhanced entrance of CANT through the air-sea interface ?

We noticed there was controversy in this part of the paragraph as the enhanced ventilation not necessarily means an increase in the invasion of $CO_2$ from the atmosphere which rise the $C_{ant}$ in the subsurface waters. Considering the path of the ENACW from the south and subsequent warming of this layer at subpolar latitude, the high $C_{ant}$ encountered in subsurface waters may be more closely related to the northward transport of $C_{ant}$-rich water than to the invasion of atmospheric $CO_2$ (as surface warming reduced its solubility). We modified this part of the paragraph as follow:

"The enhanced oxygenation of the ENACW (AOU <20 µmol kg-1 and reaching the oxygen saturation after 2014) was related with its high rates of renovation due to its path from the south (Pollard et al., 1996) and its mixing with waters moving eastward (Ellett et al., 1986). This favoured the transport subsurface waters with relatively high $C_{ant}$ content from lower latitudes into the Rockall Trough and introduced wide differences respect to adjacent deeper layers moved from the western NASPG which strength the stratification."

L651 : The name of this section « Drivers pH » should be rephrased

The previous section titled "Drivers pH" was modified by including the results from the decomposition of both pHT and $\Omega$ trends as explained above. The new title of this section (4.5) is "Processes controlling pHT and $\Omega$ trends".

L674-676 : This sentence is repeated in the conclusion. It could be removed here.

The entire conclusion was modified